# DigiData: Training and Evaluating General-Purpose Mobile Control Agents

## Abstract

AI agents capable of controlling user interfaces have the potential to transform human interaction with digital devices. To accelerate this transformation, two fundamental building blocks are essential: high-quality datasets that enable agents to achieve complex and human-relevant goals, and robust evaluation methods that allow researchers and practitioners to rapidly enhance agent performance. In this paper, we introduce DigiData, a large-scale, high-quality, diverse, multi-modal dataset designed for training mobile control agents. Unlike existing datasets, which derive goals from unstructured interactions, DigiData is meticulously constructed through comprehensive exploration of app features, resulting in greater diversity and higher goal complexity. Additionally, we present DigiData-Bench, a benchmark for evaluating mobile control agents on real-world complex tasks. We demonstrate that the commonly used step-accuracy metric falls short in reliably assessing mobile control agents and, to address this, we propose dynamic evaluation protocols and AI-powered evaluations as rigorous alternatives for agent assessment. Our contributions aim to significantly advance the development of mobile control agents, paving the way for more intuitive and effective human-device interactions.

## 1 Introduction

Mobile control agents carry out tasks on behalf of a user by interacting with a user interface on a mobile device (Rawles et al., 2024b; Wu et al., 2024). A general-purpose mobile control agent should be able to perform well in a wide range of tasks, achieving complex goals for a user. In order to be helpful, an agent should leverage the entire set of functionalities that the apps on a digital device potentially unlock. To drive progress in the field, outstanding questions remain around which data should be used to train these agents, and how they should be evaluated.

Given the potential for real-world impact, the scientific community has created a number of datasets for training mobile control agents (Rawles et al., 2024b; Li et al., 2024; Burns et al., 2021). However, existing datasets are not specifically collected for training agents to use advanced app functionalities, and lack either the depth, the diversity, or the scale to lead to general-purpose mobile control agents.

Moreover, a large portion of existing benchmarking techniques are insufficient to test these capabilities, due to inaccurate metrics or narrow task distributions. Indeed, existing work largely relies on either single-step action matching criteria (Rawles et al., 2024b; Li et al., 2024) or dynamic evaluation using a limited number of easily verifiable goals (Rawles et al., 2024a).

In this paper, we propose DigiData, a new large-scale, high-quality, diverse, multi-modal dataset.[1][2] DigiData is produced using a data generation protocol tailored to create general-purpose mobile control agents, while simultaneously ensuring high data quality. In particular, DigiData's goals are derived from a comprehensive enumeration of an app's features, allowing for deep exploration of a device's functionalities, and for unprecedented levels of goal diversity.

In addition to DigiData, we introduce an associated benchmark, DigiData-Bench. DigiData-Bench provides a curated set of diverse and interesting goals to measure the performance of agents. It supports human-assisted dynamic evaluation with a precise protocol, as well as AI-assisted dynamic evaluation with LLM judges, and allows the study or the performance of agents trained with DigiData.

---

[1]code: `https://anonymous.4open.science/r/submission-14137-5211/README.md`
[2]dataset: `https://colab.research.google.com/drive/1f_PdVvelM77G1j1PS7WgIyhjbaqQgHO-`

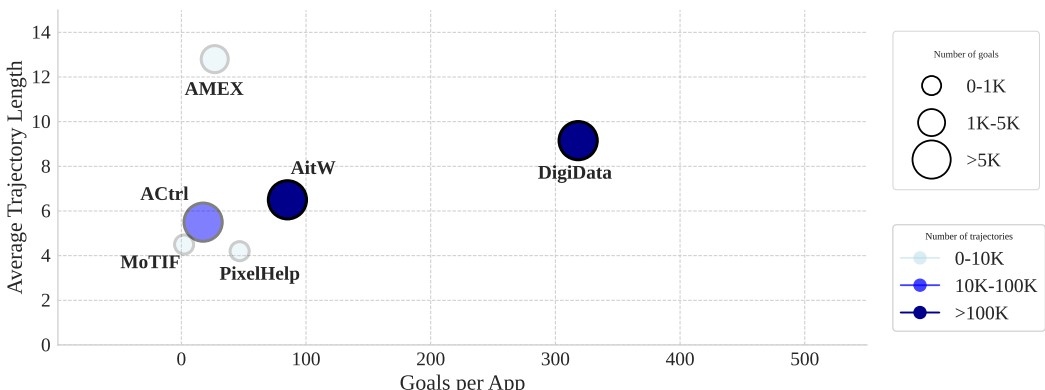

Figure 1: Visualization of different features of existing mobile control datasets. DigiData constitutes a step change in terms of goal depth, being the first large-scale dataset obtained by comprehensive exploration of the functionalities of mobile device apps.

We use DigiData for training mobile control agents, showing its usefulness for improving agents in DigiData-Bench, as well as existing benchmarks. Additionally, we measure the properties of agents trained on DigiData in terms of data and parameter scaling. Lastly, we provide the first empirical evaluation of the use of LLM judges for measuring success of mobile control agents. Overall, our paper offers new useful tools for improved training and evaluation of mobile control agents.

## 2 RELATED WORK

**Mobile Control Datasets.** To provide resources to train mobile control agents, previous work has released a number of datasets. These datasets share a common structure, offering a set of goals to be executed in mobile devices, specified in natural language. For each goal, they provide human demonstrations, consisting in a sequence of actions needed to achieve the goal, as well as a corresponding sequence of observations from the device, in the form of screenshots (Rawles et al., 2024b), UI tree descriptions (Li et al., 2024), or LLM chain of thought (Zhang et al., 2024). Previous work has proposed a number of small-scale datasets, such as MoTIF (Burns et al., 2021), PixelHelp (Li et al., 2020), UGIF (Venkatesh et al., 2022), AMEX (Chai et al., 2024), and few large-scale datasets. In particular, Android in the Wild (Rawles et al., 2024b) and AndroidControl (Li et al., 2024) are the only existing large-scale mobile control datasets, featuring tens of thousands of goals, and tens or hundreds of thousands of trajectories. More recently, there is an emerging interest for better benchmarking of mobile control agents. Notably, Rawles et al. (2024a) proposed AndroidWorld, which builds on top of existing Android emulator interfaces Toyama et al. (2021) to deliver automated evaluation of success rate in restricted settings, and concurrent work Chen et al. (2025) proposed a benchmark with text-matching-based verification.

**Computer Use.** Mobile control falls into the broader field of *computer use*, in which agents autonomously operate computers (Sager et al., 2025). A number of datasets and benchmarks have been introduced for training and evaluating agents to operate web browsers (MiniWob (Shi et al., 2017), WebVoyager (He et al., 2024), WebArena (Zhou et al., 2023), VisualWebArena (Jang et al., 2025), Mind2Web (Deng et al., 2023)) or operating systems (OSWorld (Xie et al., 2024)). Alongside benchmarks, a number of methods have been proposed for improving computer use and, specifically, mobile control, based on prompting (Yao et al., 2023; Zheng et al., 2024), supervised fine-tuning (Hong et al., 2024; Zhang et al., 2023; 2024), or reinforcement learning (Bai et al., 2024; 2025). Mobile control presents its unique challenges and peculiarities, and our work brings lessons from other types of computer use to this mobile control.

## 3 THE DIGIDATA DATASET

DigiData is a dataset designed to offer diverse and high-quality data to train mobile control agents. Differently from existing datasets, DigiData is created using a data collection protocol that attempts to comprehensively cover all app features, while simultaneously ensuring high data quality.

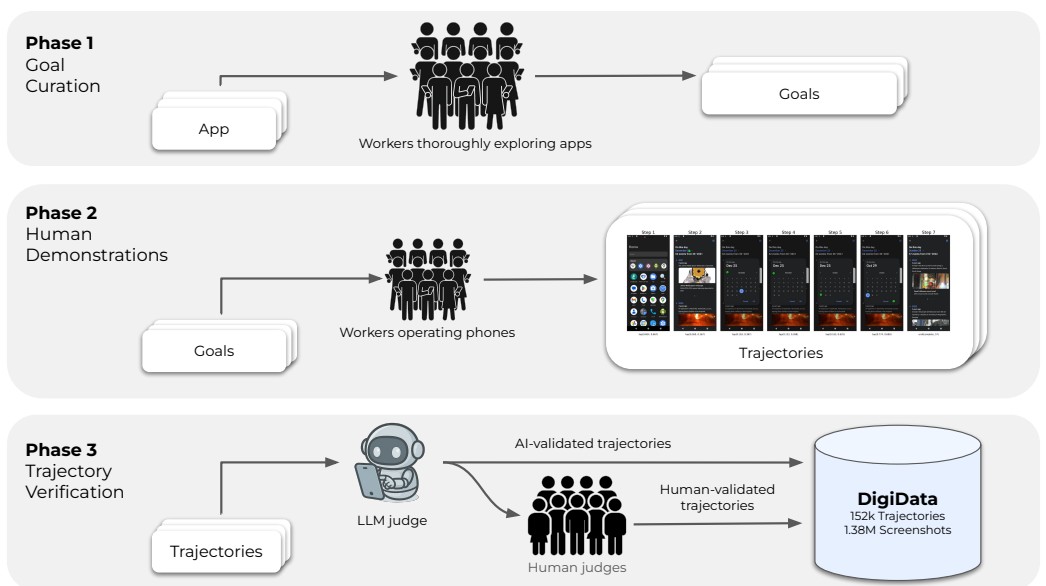

Figure 2: A representation of our data collection pipeline. For each app, our pipeline includes three phases. In the first phase (goal curation), human workers exhaustively explore the app and curate a list of goals that attempts to cover all of its features. In the second phase (demonstrations collection), human annotators create a set of demonstrations, generating trajectories that achieve the specified goals. In the third phase (trajectory verification), trajectories that do not achieve their corresponding goal are filtered out of the dataset, by a verification system based on a combination of LLMs and humans. Overall, this pipeline allows to collect in-depth and high-quality mobile control data.

## 3.1 DATA COLLECTION

Our data collection pipeline consists of three phases: goal curation, demonstrations collection, and trajectory verification. These three phases are carried out for each one of the apps included in our dataset. Figure 2 visually represents the data collection pipeline.

**Goal Curation.** To collect a dataset of high quality demonstrations, we start by generating high quality goals. We instruct a small group of trained annotators to exhaustively explore an application's features by navigating the application's screens and menus to curate a set goals related to those features. To limit the maximum complexity of a goal while simultaneously covering deeper features in the app, we generate a portion of the goals as part of a sequence of goals.

**Demonstrations Collection.** Our data collection system exists as a custom Android application that can be deployed on an un-tethered physical or emulated device. The app automatically selects a goal, records the screen state and demonstrated action the annotator takes to complete the selected goal. Upon completion, the annotator reviews the recorded trajectory in the app before uploading to our secured servers. We employ internal contractors that use real physical Android devices as well as crowd-sourced online annotators with access to our secure emulation environment.

**Trajectory Verification.** To ensure the quality of DigiData we employ a combination of AI-based and human-based verification methods post collection. Given a trajectory produced by our annotation platform, we utilize an LLM judge tuned to have a low false accept rate to automatically evaluate trajectory success. Given the risk of false negatives from the LLM model, we send trajectories deemed as unsuccessful to human annotators, and add them to the dataset if either LLM or human annotator judge it as successful. 5.3% of the raw trajectories get filtered out by this procedure.

## 3.2 DATASET FEATURES

We now compare DigiData to existing datasets for mobile control, highlighting its properties and advantages. Table 1 summarizes the main features of existing large-scale datasets and shows how the

| Dataset | # Trajectories | # Goals | # Apps | Trajectory length | Diverse modalities | Goals per app | Diversity score |
|---|---|---|---|---|---|---|---|
| AitW | 715k | 30,378 | 357 | 6.5 | $\times$ | 85 | 0.35 |
| AndroidControl | 15k | 14,548 | 833 | 5.5 | $\checkmark$ | 17 | 0.43 |
| DigiData | 152k | 8,275 | 26 | 9.2 | $\checkmark$ | 318 | 0.45 |

Table 1: Statistics on large-scale mobile control datasets. DigiData features exploration of each app at unprecedented scale, as visible in its average trajectory length, goals per app, and diversity score.

features of DigiData make it a fundamental advancement with respect to existing datasets, which are potentially complementary to it.

**Dataset Size and Quality.** DigiData features 8,275 unique goals across 26 Android apps. In total, our dataset consists of 152,000 trajectories, positioning it as the second largest dataset for mobile control after AitW. To compare the quality of our dataset to the one of AitW, we sent 20k trajectories randomly sampled from AitW through our human trajectory verification system. According to our human annotators, only 84% of AitW trajectories achieves the prescribed goal, compared to 94.6% of DigiData before our verification step and 100% after verification: this makes it easier to train an agent using DigiData, and can be seen as evidence that DigiData is a state-of-the art mobile control dataset in terms of data quality.

**Modalities.** DigiData is the first dataset of its scale to feature multiple input modalities. In particular, in addition to the screenshot, we include two additional input modalities for each step in a trajectory, to aid future research on observation processing and encoding for mobile control agents. First, we provide a UI Tree, the underlying Android OS accessibility tree generated at the time of the screenshot. Second, we offer Chain-of-Thought data (Wei et al., 2022) generated via Llama 4 (Team, 2025a) (seen next paragraph), which can be used as training data to increase explainability in mobile control agents. These two input types were provided by previous datasets of significantly smaller size (Li et al., 2024; Zhang et al., 2024), but no public dataset provides them at the scale of DigiData and with permissive licensing.

**Chain-Of-Thought Data Generation.** To foster agent actions' explainability and potentially improve performance of agents trained with DigiData, we provide additional LLM-generated annotations (Wei et al., 2022) for each time step in the dataset. We used Llama 4 (Team, 2025a) to produce supplementary descriptions from parsed versions of the UI tree and the screenshot at each step. These additional annotations provided include the screen description (*observation*), the human-understandable description of the action (*action short*), rationale for enacting the action (*thought*), and description of the resulting screen by enacting the action (*expected UI change*).

**Goal Complexity.** One of the main differentiating features of DigiData compared to existing datasets is the complexity and richness of its goals. Goals in DigiData require approximately 9.2 steps on average to be achieved by a human demonstrator, roughly 50% or more than existing datasets. The additional complexity is the result of the goal curation protocol of our data collection pipeline, which generates goals tasking annotators to explore and leverage deep app features. This manifests in an unprecedented average number of goals for each app compared to existing datasets, with 318 goals per app compared to 85 of AitW and 17 of AndroidControl. We believe training agents to use advanced app functionalities paves the way to potentially very large practical impact, since these advanced app functionalities are often unfamiliar to most digital device users (which may require help in using them) and are often only supported via user interfaces (e.g., being excluded from APIs).

**Diversity.** As shown in Figure 8, DigiData features a set of diverse apps, and app categories, and a balanced amount of goals for each one of them. This sets DigiData apart from datasets such as AitW, which are characterized by a more imbalanced set of goals (e.g., with a large portion of the goals being related to e-commerce apps). These goals are not only well-balanced across apps, but also highly diverse, due to the process by which they were collected. Since advanced functionalities of each app can vary wildly one from the other, comprehensive exploration of the app's features yields higher goal diversity. To quantify this, we encode the natural language goal instructions included in the DigiData, AitW, and AndroidControl datasets and compute the pairwise cosine "distances" between all the resulting vectors as $d(\mathbf{x}, \mathbf{x}') = \frac{1}{2} - \left( \frac{\mathbf{x} \cdot \mathbf{x}'}{2\|\mathbf{x}\| \cdot \|\mathbf{x}'\|} \right)$. The resulting distances lie between 0 and 1, with higher values denoting greater vector diversity. Figure 3b shows a histogram of this

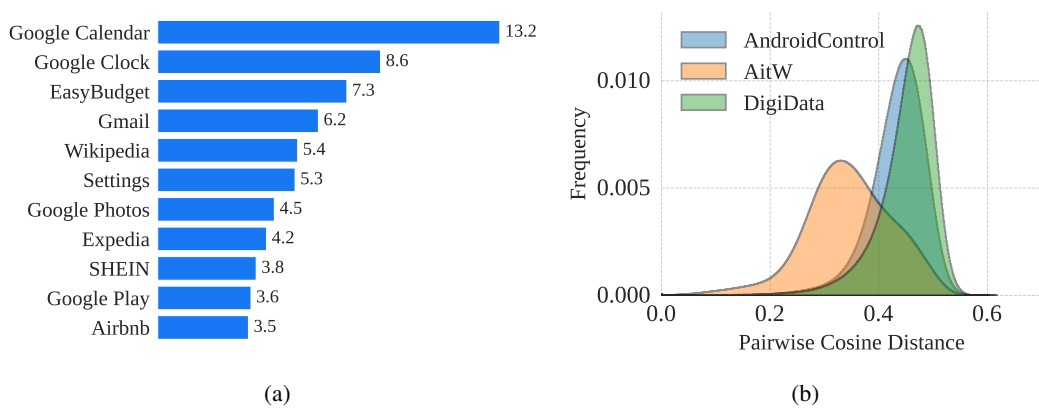

(a)                (b)

Figure 3: (a) Percentage of data distribution for DigiData's top apps. The dataset presents no major imbalance towards specific apps. (b) Comparison of distribution of pairwise cosine distances across datasets. DigiData exhibits the largest degree of goal diversity, especially compared to AitW.

pairwise metrics across the three datasets: the results show that DigiData features higher levels of goal diversity, especially compared to AitW, with high levels of pairwise distance being frequent across the dataset. The last column of Table 1 also shows the average metric across the dataset, which constitutes a *diversity score* for the sets of goals in the different datasets.

## 4    THE DIGIDATA-BENCH BENCHMARK

Alongside DigiData, we propose DigiData-Bench, a benchmark to evaluate mobile control agents. DigiData-Bench serves a double purpose: on the one hand, it directly allows carefully evaluation of agents trained on DigiData; on the other hand, it provides a reproducible protocol and platform for human-assisted or AI-automated evaluation of mobile control agents in general.

**Apps and Goals.** DigiData-Bench is a benchmark constituted by 309 goals spread across 37 Android apps and 8 app categories. We designed its goals to capture a wide range of use cases, similarly to DigiData, but also selected them for being particularly interesting and not directly included in DigiData's goals. For the variety of real-world apps and tasks it features and the generality of its evaluation method, evaluations provided by DigiData-Bench are complementary to the ad-hoc verification on open-source local apps provided by existing work (Rawles et al., 2024a).

**App Novelty Categories.** We identified three novelty categories with respect to the apps in DigiData. This allows to study the generalization properties of trained agents. We categorize goals into a *seen* group, when the corresponding app is included in the DigiData training set, a *familiar* group, when the corresponding app is not included the training set but an app of the same category (e.g., e-commerce) is, and a *novel* group, when there is no app of the same category in the training set. The familiar and novel groups can be used to test generalization capability of the out of domain apps.

### 4.1    HUMAN-ASSISTED DYNAMIC EVALUATION

To create a reproducible setup to evaluate an agent for each one of the goals in DigiData-Bench, we provide a description of a goal-specific evaluation protocol. This description serves as a guide for human workers, that create the setup for the agent to achieve the goal, monitor the agent, and detect whether the trajectory generated by the agent was successful.

**State Initialization.** To properly evaluate whether an agent has achieved a goal or not, prerequisites should be in place for such goal to meaningfully achieved. For instance, if the goal consists of finding or deleting an item inside of a specific app, that item should exist, or otherwise the goal would be trivial. For each one of the goals, we provide *state initialization instructions*, which human workers should execute before running the agent. These instructions are an attempt not only to ensure that the goal can be meaningfully achieved, but also to guarantee reproducibility across evaluation runs (e.g., by doing ad-hoc re-initialization of app functionalities).

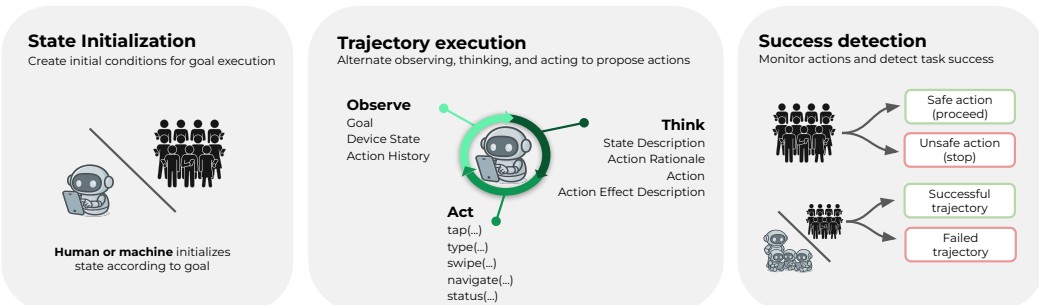

Figure 4: A visual representation of the evaluations on DigiData-Bench. Following a goal-specific protocol, a human worker initializes the app for meaningfully achieve the goal, monitors the agent executing actions and detects whether the agent was successful in achieving the goal.

**Trajectory Execution.** After having initialized the state, human operators start the agent interactions with the app required to achieve the goal. Because we evaluate against real world applications in the wild, we have human operators monitor the agent's actions, following a strict set of rules for allowed operations (see supplementary material for full set of operator instructions), stopping the agents from taking any unsafe actions. On average the agent was stopped for $0.5\%$ of trajectories with the vast majority of flagged action related to enabling permissions without the users permission.

**Success Detection.** When the agent generates a termination action, generates an unsafe or invalid action, or enough steps have elapsed, the trajectory is terminated. Then the human operator is asked to judge whether the trajectory successfully achieves the goal or not. Averaged across runs and goals, this provides a success rate metric to evaluate agents. Given the complexity of some of the apps and the complexities of mobile control, detecting whether a trajectory was successful or not can sometimes be non-trivial even for careful human workers. For this reason, our protocol includes additional *success detection instructions* that help human workers in their verification tasks.

## 4.2 AI-ASSISTED DYNAMIC EVALUATION

Our protocol for human-assisted dynamic evaluations requires a human to operate a digital device to initialize the state, monitor the trajectory execution, and to detect success. While this is the most general protocol supporting the evaluation of a mobile control agent on potentially any application for potentially any goal, human involvement can add an overhead to agent evaluations. For this reason, we propose an AI-assisted benchmark derived from DigiData-Bench, that we name DigiData-Bench-Auto, which is and end-to-end testing suite (protocol) that automates inference and evaluation via an LLM judge on a tethered Android device.

**Goal Selection.** We select, among the goals of DigiData-Bench, the ones that are more easily reproduced in an automated way. These are the goals that require no manual state initialization, no login or location information, and have low risk of dangerous side effects, in order to be meaningfully verified. The goals are similar in spirit to recent benchmarks for web navigation, such as GAIA (Mialon et al., 2023) and WebVoyager (He et al., 2024), but specifically tailored to the mobile control case. The total number of goals we select amounts to roughly half of DigiData-Bench's goals.

**Evaluation Protocol.** To automatically evaluate the success of a trajectory, we propose the use of LLM judges. Given a goal and a trajectory produced by an agent, an LLM judge classifies whether it successfully achieved the goal. We use LLM judges relying on both the screenshot and UI tree. To increase the reproducibility of agent evaluation, we automatically close and reset apps and open the correct app, before running the agent and obtaining a success score.

## 4.3 OFFLINE EVALUATION

To complement dynamic evaluation judging overall success of an agent in a trajectory, we also utilize offline evaluation in the form of per-screenshot action matching as done in previous work (Rawles et al., 2024b) referred to as *step-accuracy* in this work. Despite DigiData-Bench's dynamic evaluations provide a stronger signal around an agent's performance, it can be valuable to have a proxy offline

metric. To perform this offline evaluation, we have collected a test set of human demonstrations generated with the same protocol employed for DigiData and the same goals used in DigiData-Bench.

## 5 EXPERIMENTS

### 5.1 EVALUATION SETUP

To evaluate the effectiveness of our DigiData and DigiData-Bench we train a mobile control agent that given a goal, history of previous actions and current state, is able to predict the next action along with a description of the action, current state and estimated next state. Additional experimental details (including prompts using for training and evaluation) are reported in appendix.

**Models and Data.** We fine-tune Perception Language Model (PLM) (Cho et al., 2025), as it has high performance across a number of vision-language tasks and it provides a range of model sizes that enable fast experimentation. Based on the pre-trained checkpoint of PLM, we run supervised finetuning (SFT) using a data mixture that includes four datasets (three mobile control datasets and one general VQA dataset): our DigiData, AitW (Rawles et al., 2024b), low-level and high-level instructions from AndroidControl (Li et al., 2024), and Cauldron (Laurençon et al., 2024). We add AitW and AndroidControl to validate agent performance in the realistic use-case in which multiple datasets are combined, and Cauldron to prevent catastrophic forgetting of pre-trained ability of general visual understanding.

**Agent Training.** To train and evaluate on the different mobile control datasets, we use a unified action space and use the same action only training prompt for all three datasets. Since DigiData also provides Chain-of-thought (COT), we create a second training prompt eliciting COT, to use in addition to a default training prompt that only elicits actions. In the COT setup, the model not only predicts the parameterized action but also describes the current state, reasoning for the predicted action, description of the action and the expected resulting state. COT can not only increase agent performance but also provides explainability for an agent's actions.

**LLM Judge Training.** We utilize an LLM judge for both automatic verification during data collection for DigiData and automated evaluation in DigiData-Bench. Our judge works in two stages. In the first stage, each step consisting of the starting state, action and resulting state, is transcribed into text descriptions using a *step summarization* module. Each state comprises a screenshot image and UI elements extracted from the UI tree. In the second stage, the text descriptions of all steps, along with the initial and final states of the trajectory are provided as input to the LLM judge to evaluate goal achievement. We use GPT4o, Llama 4, Llama 4 judge, and a fine-tuned Llama 4. For fine-tuning, we collect a training dataset containing 7057 failed model-generated trajectories and 2034 successful ones. We first generate step summarization and reasoning via our first stage zero-shot model. Then we fine-tune our second stage LLM judge using this step summarization as input and the generated reasoning and binary ground-truth human judgments as target output. We evaluate both the zero-shot and fine-tuned LLM judge models over a test set of balanced successful and unsuccessful model generated trajectories.

**Baselines.** We consider two popular foundation models as zero-shot baselines: QWen2.5VL (Team, 2025b) and GPT4o. We choose GPT4o since it demonstrates ability to control mobile devices with proper prompting in (Li et al., 2024), and QWen2.5VL since Team (2025b) reports results on mobile control benchmarks. Implementation details, including the prompt, are in the supplemental. We also consider step-accuracy reported by AitW and AndroidControl on their test set, using the dataset name as baseline name, as well as the one reported by Hong et al. (2024).

### 5.2 RESULTS

Table 2 shows both the offline step-accuracy metric on three benchmarks as well as the online task success rate on our DigiData-Bench. We compare against a number of SotA models as well as a baseline model trained on just public data without our DigiData dataset to show the benefit of DigiData. Our model significantly outperforms both our baseline and industry leading propriety (GPT4o) and closed data (Qwen2.5VL) models on our dynamic DigiData-Bench. We observe that the model trained with COT data gives a significant improvement to our model performance hitting

|  | Step-accuracy | | | | Success Rate (DigiData-Bench) | | | | |
|---|---|---|---|---|---|---|---|---|---|
|  | DigiData-Bench | AitW | ACtrl(H) | ACtrl(L) | All | Seen | Familiar | Novel | All (LLM) |
| GPT4o | 40.0 | - | - | - | 27.8 | 26.7 | 33.3 | 26.5 | 38.0 |
| Qwen2.5VL Team (2025b) | 49.2 | - | - | - | 39.2 | 37.9 | 42.6 | 40.8 | 46.2 |
| AitW Rawles et al. (2024b) | - | 73.1 | - | - | - | - | - | - | - |
| AndroidControl Li et al. (2024) | - | - | **71.5** | **86.6** | - | - | - | - | - |
| CogAgent Hong et al. (2024) | - | 76.8 | - | - | - | - | - | - | - |
| Ours 1B | 67.6 | 77.4 | 64.1 | 78.5 | 35.0 | 38.4 | 37.0 | 18.4 | 37.7 |
| Ours 3B | 70.7 | 78.0 | 65.3 | 82.1 | 44.3 | 49.5 | **46.3** | 20.4 | 49.8 |
| Ours 8B | 70.7 | 78.1 | 64.1 | 81.4 | 42.1 | 45.2 | 44.4 | 26.5 | 48.5 |
| Ours 8B COT | **72.8** | **78.7** | 63.8 | 77.8 | **47.3** | **51.0** | 42.6 | **36.7** | **53.6** |

Table 2: Comparison of step-accuracy across different datasets and success rate on DigiData-Bench. ACtrl(H) and ACtrl(L) refer to high-level and low-level tasks from Android Control's Li et al. (2024) test set, while success rate on DigiData-Bench is reported on three app novelty subsets. All (LLM) reports success rate from GPT4o as a judge model. Ours 8B COT has the highest success rate on DigiData-Bench from both human evals and LLM judge showing the effectiveness of our DigiData and the synthetic COT data.

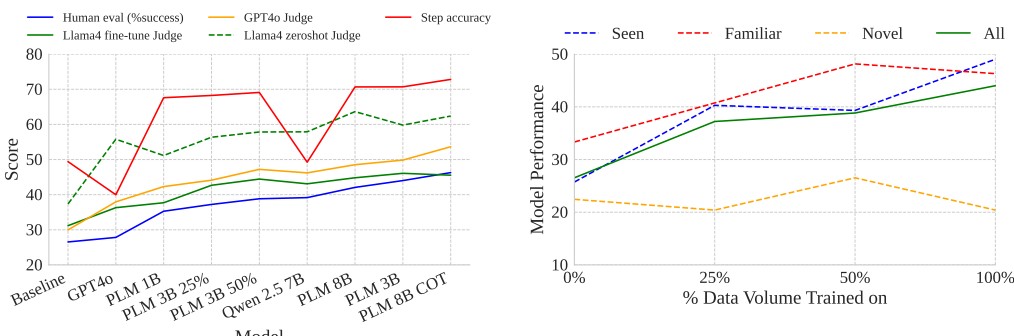

Figure 5: **(Left)** Comparison of evaluation metrics across a number of models. Rankings provided by step accuracy are generally not reliable. **(Right)** Success rate by app category. Training agents with more data significantly improves performance on seen and familiar apps, but not on novel apps.

47% task success rate. This shows that COT data not only improves explainability, but also agent performance.

**Online Evaluations Matter.** Figure 5 (left) shows the performance on DigiData-Bench for both static and dynamic evaluation protocols across a number of models ordered by human evaluation of task success rate. While there is strong correlation between all three judge models and the human evaluation, step accuracy is not necessarily a good predictor of the ordering of agent models with respect to the human-evaluated performance. Thus, even though step accuracy could be a convenient proxy measure to use during model development, task success rate is needed to accurately measure an agent's capability. This can also be observed in Table 2, in which our 3B and 8B models have close to identical step accuracy on both DigiData-Bench and AitW, but significantly different performance on DigiData-Bench success rate.

**Human Evaluation.** Online evaluation is only valuable as a metric if it has relatively low measurement error. To measure the variance of our human evaluation we ran evaluation for our 3B model through human evaluation 3 times. We found an STD of $\pm 0.4\%$ showing our metric has relatively low measurement error. We also measured human expert performance at 90.1% task success rate, highlighting the large gap remaining between mobile control agents and humans.

**LLM Judges as Proxy Dynamic Evaluators.** For producing most of the results in Table 2, we employed humans to watch and evaluate agents as they interact with DigiData-Bench tasks. To investigate how LLMs can automate this process, we evaluate LLM judges for the task of detecting success given a goal. In Figure 5 (left), we show that, even though the LLM does not have full agreement with human judges, LLM judge evaluations are generally well-correlated with agent rankings, as clear in the visual trend. Leveraging an LLM judge enables running a set of relevant DigiData-Bench goals in an automated way, reducing the need to only rely on step-accuracy for model development.

| Agent Judge | Accuracy | Precision | Recall | NPV | TNR | Kendall Rank Correlation with human judgement |
|---|---|---|---|---|---|---|
| Llama 4 Scout | 0.82 | 0.73 | **0.89** | **0.91** | 0.77 | 0.83 |
| Llama 4 Scout (fine-tuned) | 0.87 | **0.88** | 0.79 | 0.86 | **0.92** | 0.89 |
| GPT4o | **0.89** | 0.87 | 0.86 | 0.9 | 0.9 | **0.94** |
| Step accuracy | - | - | - | - | - | 0.72 |

Table 3: Comparison of agent judges, evaluated using six metrics: Accuracy, Precision, Recall, Negative Predictive Value (NPV), True Negative Rate (TNR), and correlation with human judgement. The open weight Llama 4 fine-tuned on a small set of data is able to reach comparable performance to GPT4o. Agent evaluation by LLM judges generally agrees with human judgement, providing a significantly stronger signal to agent quality compared to step accuracy.

**Evaluating LLM Judges.** With the goal of carefully evaluating LLM judges for mobile control, we construct a balanced test set of successful and unsuccessfully model generated trajectories and compute common binary classification metrics. Table 3 shows the performance of a number of different zero-shot models, along with a fine-tuned version of Llama 4. GPT4o is the strongest LLM judge for mobile control, but relatively fast to run open source models can reach close performance by fine-tuning on just a few thousand samples. In particular, the fine-tuned Llama 4 judge is able to come close to match its performance; it even slightly outperforms it in terms of TNR, which is a key metric for flagging unfit trajectories in the trajectory verification phase of DigiData's data collection. Table 3 also reports the rank correlation between rankings according to human judgement around models from Table 2 and ranking according to various judging metrics (including step accuracy). LLM judges outperform step accuracy by a large margin in providing accurate agent evaluation.

**Data Scaling.** To assess the impact of training agents with increasing amounts of data, we trained four models with varying data volumes. Figure 5 (right) shows the effect of including different amounts of DigiData in the training mixture. Overall performance continues to improve with more data, showing the benefits of DigiData. However, performance does not significantly improve for *novel* apps. This points to inherent limitations of approaches based on supervised fine-tuning, which can exhibit lack of generalization compared to reinforcement learning (Chu et al., 2025). We believe this evidence could inspire future work to focus on training agents with reinforcement learning.

## 6 CONCLUSION

In this paper, we presented DigiData, a dataset for training mobile control agents. DigiData is a large-scale, high-quality, diverse dataset obtained by comprehensive exploration of the functionalities of mobile apps. Alongside DigiData, we proposed DigiData-Bench, based on human- and AI-assisted agent evaluation protocols. We trained mobile control agents using the dataset and evaluated them on DigiData-Bench, demonstrating that DigiData allows agents to achieve strong performance according to both accuracy and success metrics. Finally, we evaluated LLM judges of mobile control agents, showing that the promising performance of existing and fine-tuned models has the potential to enable even further automation of mobile control agent evaluation. We believe DigiData is a concrete step to better train and evaluate general-purpose mobile control agents.

**Reproducibility** We strongly believe our work is fully reproducible. We provide all training and test data along with detailed instructions and code for setting up and running dynamic evaluations. Our model is trained directly using the PLM Cho et al. (2025) code base. Our training recipes and prompts are provided in the appendix.

**Limitations and Future Work.** While our proposed dataset, DigiData, significantly advances the development of mobile control agents, there are limitations to be addressed in future work. Notably, DigiData's is limited to set of 26 mobile apps. The number of apps and scale of DigiData, although substantial, may not match the vast diversity of real-world applications. Furthermore, its generalizability to more general mobile control such as computer-based interfaces remains unexplored. Our evaluation protocols, while robust, rely on human-assisted or AI-assisted evaluation, which may introduce biases or limitations. To further enhance the field, future research should focus on expanding DigiData to include a broader range of applications, exploring transfer learning capabilities to novel apps, and adapting our methods to accommodate computer-based interfaces, ultimately paving the way for more comprehensive and effective mobile control agents.

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

## A    LLM USAGE STATEMENT

We used a large language model (LLM) to help polish the conclusion section of this paper. No other significant use of LLMs occurred during the preparation of this manuscript.

## B    TRAJECTORY VISUALIZATION

Figure 6 shows some trajectories actuated by the mobile agent model. We observe the model trained on DigiData can navigate a mobile device and complete complicated task requiring multiple steps.

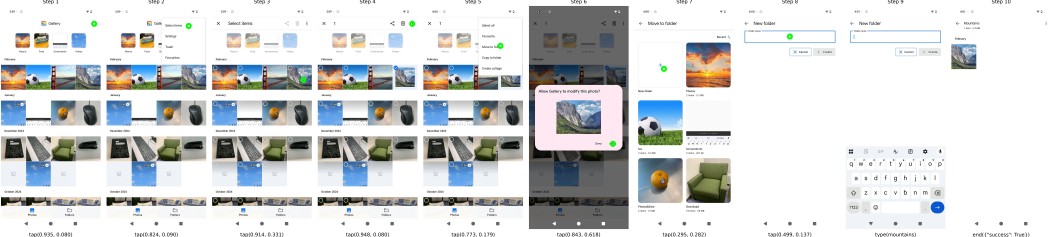

(a) Goal: Using Gallery (Google) app, create a new folder named "Mountains" and move the most recently taken photo of a mountain landscape into it.

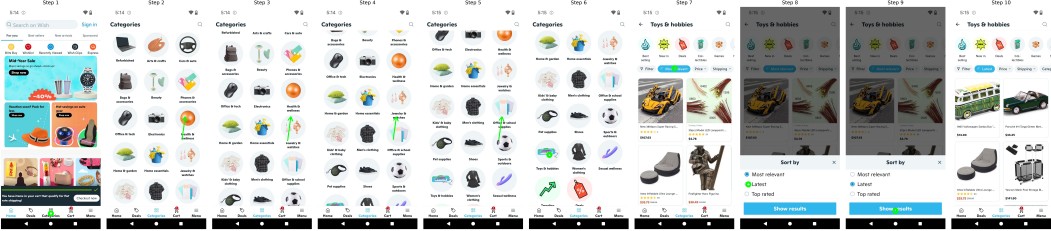

(b) Goal: Using Wish app, navigate to products in the Toys category and sort by "latest".

Figure 6: Visualization of the output of our approach on two goals from the DigiData-Bench. Tapping and swiping actions are visualized for each step.

## C    BENCHMARK DETAILS

**Diversity of Apps and Goals.** DigiData-Bench apps and goals have a wide distribution across 8 app categories: Communications, Food, Management, Media, Navigation, Restricted Search, Shopping and Misfit. Detailed goal counts per app and app category mapping can be found in Figure 7.

**Step accuracy computation.**   Two actions are considered a match if their action types are identical and their parameters fall within a specified threshold. For instance, two tap actions are considered equivalent if they are within a certain screen distance from each other. At each step, the agent is given the goal and one of the screenshots from the demonstration, and asked to produce an appropriate action at that step. The prediction is considered correct if the action matches and wrong otherwise.

**Goal similarity categories.** We classified the goals into two similarity categories concerning the objectives within the *seen* app groups. Goals are categorized as *Similar* when goals exist in the same app in DigiData that require almost identical execution. It includes goals with different parameterizations of the same goal template or different rephrasings of the same goal. *Unseen* goals refer to no similar goals exist in the same app in DigiData. The *Unseen* category can be used to test generalization capability of the model.

**Goal intuitiveness categories.** We further classified each goal based on the ease of its execution that can be used to evaluate model's search capability. *Intuitive* goals can be easily performed by an average user without prior knowledge of the app's internal mechanisms. Each subsequent step should be clear and self-explanatory. For instance, "Search for dog videos on YouTube". *Non-intuitive* goals are not immediately obvious and may require some degree of searching or exploration within the app.

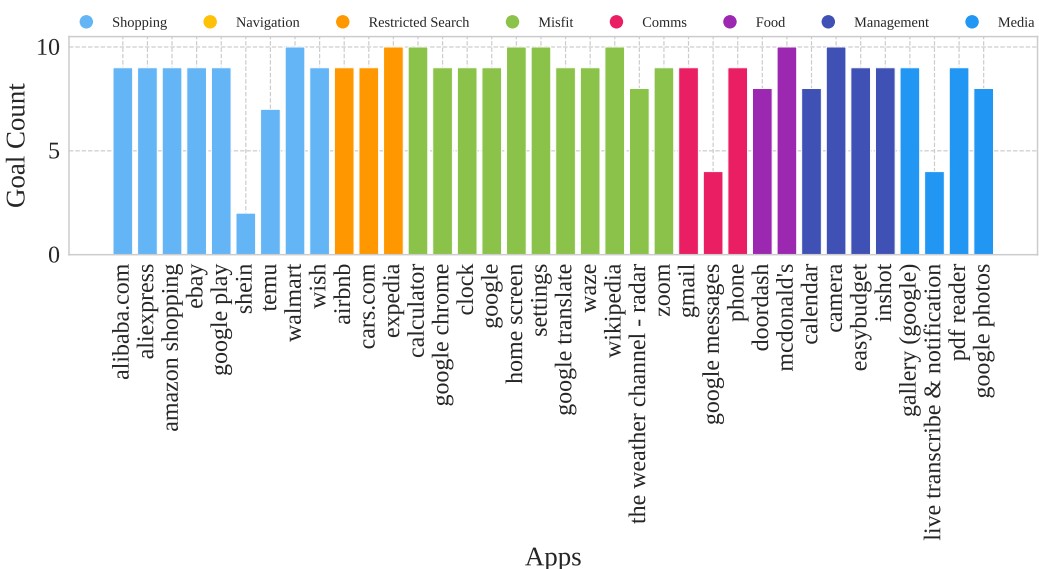

Figure 7: Goal Count per App and App Category Distribution

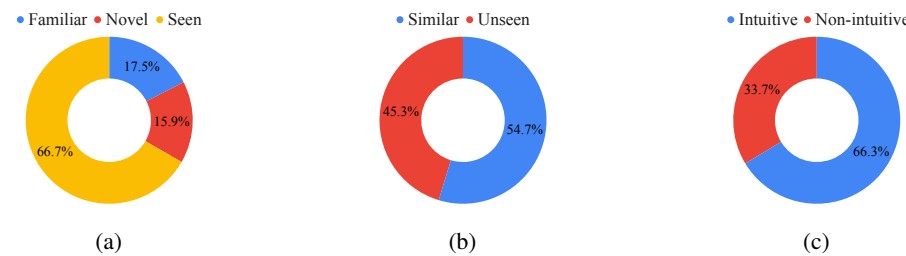

Figure 8: (a) App novelty distribution of DigiData-Bench. (b) Goal similarity distribution of DigiData-Bench. (c) Goal intuitiveness distribution of DigiData-Bench

An average user might need to consult external resources to understand how to accomplish them. For example, "Disable stable volume for videos on YouTube." It is important to note that non-intuitive objectives do not necessarily involve longer execution paths, and the reverse is also true.

## D    LIMITATIONS OF STEP ACCURACY EVALUATION

The capability of a device control agent to navigate Android phones involves not only completing tasks but also understanding that multiple routes can lead to the same outcome. For example, when attempting to display a specific item, a user might either directly utilize the search bar or navigate through categories, accessing multiple directory levels to find it. This phenomenon has significant implications for evaluating the performance of device control agents. In step accuracy evaluations, only one path is typically considered correct, potentially overlooking other valid routes. In contrast, evaluating the trajectory-level task success rate offers a more comprehensive assessment of the AI agent's performance, as it accounts for the various routes that can lead to successful task completion. As illustrated in Figure 9, three distinct navigation modes - direct search, scrolling through menus, and using an alphabetic index - can all achieve the same goal, "Change the shipping location to Dominican Republic", highlighting the importance of considering multiple navigation paths when evaluating AI agent performance.

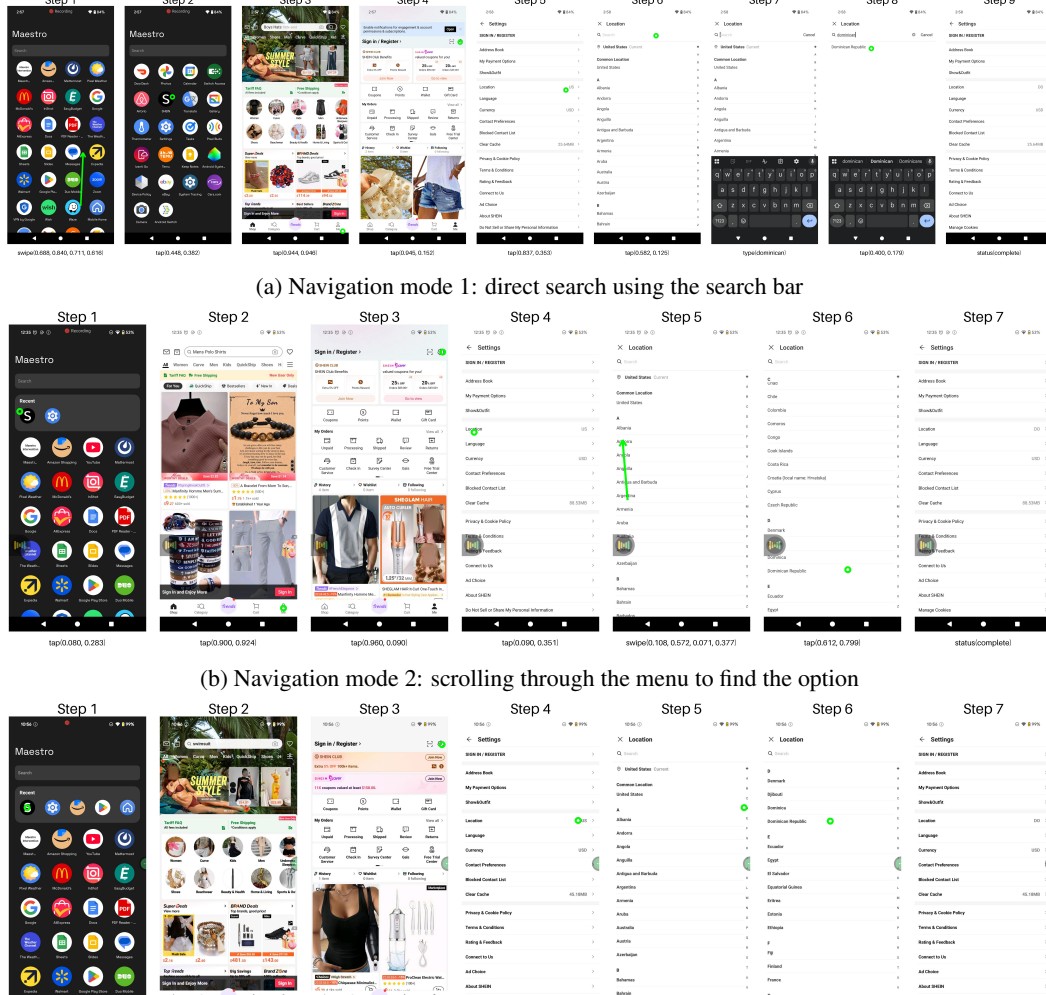

(a) Navigation mode 1: direct search using the search bar

(b) Navigation mode 2: scrolling through the menu to find the option

(c) Navigation mode 3: utilizing the alphabetic index bar to jump directly to the relevant group and select

Figure 9: Three distinct navigation modes for achieving the same goal: "Change the shipping location to Dominican Republic".

# E    PRELIMINARY INSIGHTS INTO THE EFFECTS OF TASK COMPLEXITY

In light of the findings from human evaluations regarding task success rates, we conducted a preliminary investigation to explore the relationship between task complexity and model performance. Given the challenges associated with directly quantifying task complexity, we employed a proxy metric: the average number of steps required for users to successfully complete each task. Although this approach has its limitations—primarily due to the imperfect correlation between step count and task complexity—it nonetheless offers initial insights into this domain. Figure 10 illustrates the outcomes of this investigation. The line chart represents the task success rate for each model, while the bar histogram depicts the total number of tasks corresponding to each step range. Our results indicate a discernible trend: task success rates tend to decrease as the number of steps increases, suggesting that model performance diminishes with rising task complexity.

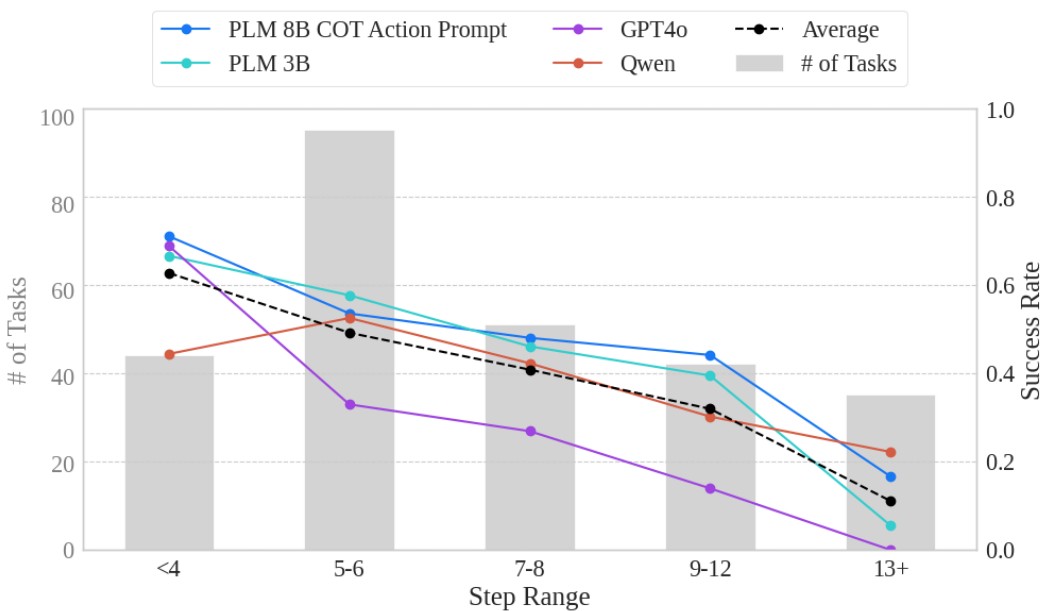

Figure 10: Task Success Rate vs. Task Step Count

| Dataset | Volume | Epochs | Sampling propability |
|---|---|---|---|
| DigiData | 1.38M | 1.05 | 11.3% |
| DigiData-CoT | 1.38M | 2.10 | 22.7% |
| Android-in-the-wild | 5.5M | 0.64 | 27.9% |
| Android Control (low-level) | 32K | 4.12 | 1.0% |
| Android Control (high-level) | 32K | 12.4 | 3.1% |
| Cauldron | 3.6M | 1.21 | 34.0% |

Table 4: Our data mixture of supervised finetuning (Ours 8B CoT), including data volume (number of images), estimated epochs on each dataset and the sampling probability. We do not make a full pass of Android-in-the-wild, since we find the step accuracy on Android-in-the-wild has already saturated. We made multiple pass of Android Control, since it is a small dataset.

# F EXPERIMENTAL DETAILS

## F.1 SUPERVISED FINETUNING

**Data mixture.** We include four datasets in the final data mixture of the supervised finetuning stage: Android-in-the-wild, Android Control, DigiData, and Cauldron. A summary is shown in Table 4. For Android-in-the-wild, and Android Control, we construct action history of three pass steps, and include it in the text prompt. The detailed text prompt is shown in Prompt 1. We throw away trajectories with incomplete steps. For DigiData, we construct action history in a similar way. For Cauldron, we remove QA pairs with interleaving images since our agent focuses on a single screenshot.

**Implementation details.** We use 128 NVIDIA A100 to train the model, and 32 NVIDIA A100 for evaluation. The learning rate is 2e-5. The batch size is 2 samples per gpu. We train the model for 60k iterations and pick up the best model for human evaluation according to the step accuracy. During fine-tuning, we fine-tune the whole model, including the vision encoder and adaptation layers. The size of each image tile is 448x448, and we limit each image has 4 tiles at most. For all evaluations, we use greedy decoding.

### F.2    BASELINES.

We compare our agent model with two popular foundation models: GPT4o and QWen2.5VL. Since they are general-purpose multimodal LLMs, we need proper prompting to make them work on Device control agent tasks. Here we include implementation details of these two models.

**GPT4o.** We follow the success of Android World Rawles et al. (2024a) and SeeAct Zheng et al. (2024) to use GPT4o to navigate web and Android UI. We adopt their prompt for our benchmark. Notably, we find GPT4o has difficulty generating correct coordinates despite understanding the screenshot. Therefore, we add a simplified XML layout to help it. The extraction of simplified XML is the same as the one we use in data generation and LLM judge. The detailed prompt is shown in Prompt 6.

**QWen2.5VL.** We run QWen2.5VL 7B using the provided mobile use prompt[3]. There was some discrepancy of action space. Therefore we had to remove some of their "tools" and adapt others to align with our action space. The detailed prompt is shown in Prompt 7

### F.3    LLM JUDGE

We curate a test set comprising 327 goals. For each goal, human judges evaluate trajectories sampled from our agent model. We randomly select one successful trajectory and one failed trajectory for each goal. However, some goals are challenging, and we were unable to collect any successful model-generated trajectories for them. Ultimately, our test set includes 229 successful trajectories and 327 failed trajectories.

## G    EVALUATING LLM JUDGES OF DEVICE CONTROL

LLM judges are gaining increasing popularity for training and evaluating AI agents (Klissarov et al., 2024; Lù et al., 2025). DigiData's data collection protocol relies on LLM judges to ensure data quality, and LLM judges are also leveraged in DigiData-Bench to automate agent evaluation. Thus, we provide an evaluation of LLM judges of device control. We curate an ad-hoc test set that includes 229 successful trajectories and 327 failed trajectories.

**LLM judge architecture.** To reduce context length requirements while maintaining performance, we adopt multi-step judging approach. Instead of inputting all the screenshots and XML data from each step into the LLM judge at once, we processes the data in stages. First, each step, comprising the action taken and the observed screenshots and XML data before and after the action, is transcribed into text descriptions using a *step summarization* module (Prompt 3). Then, we concatenate the text descriptions of all steps, along with the initial and final screenshots and XML data of the trajectory, and input them into the actual LLM judge for reasoning and evaluating goal achievement (Prompt 5). In addition to utilizing a zero-shot LLM judge, we also explore fine-tuning an LLM judge for the task. We generate a training dataset by collecting 7057 failed model-generated trajectories and 2034 successful ones, and using a zero-shot model to provide both step summarization and reasoning. We then fine-tune the LLM judge using the goal and step summarization to produce the generated reasoning and the binary ground-truth human judgments.

**Results.** We evaluate different LLM judges, including Llama 3 (Grattafiori et al., 2024), Llama 4 Scout (Team, 2025a), GPT4o and a fine-tuned Llama4 Scout. We use five key metrics for binary classification: Accuracy, Precision, Recall, Negative Predictive Value (NPV), and True Negative Rate (TNR). The results, reported in Table 3 in the main paper, show that, using our proposed modular strategy, LLMs generally obtain promising, albeit not perfect, performance in the task of judging device control trajectories. Moreover, while GPT4o generally offers the strongest performance, fine-tuning accessible open weight models on a few thousands samples only can reach similar levels of precision, showing potential for future research on fine-tuning strategies for LLM judges of device control.

---

[3]`https://github.com/QwenLM/Qwen2.5-VL/blob/main/cookbooks/mobile_agent.ipynb`

| Field | Type | Description |
|---|---|---|
| episode_id | String | Each trajectory has a unique episode id |
| step_id | Integer | Zero-indexed, current position in the episode |
| episode_len | Integer | The total length of the episode, not reflecting missing steps |
| app | String | The Android app necessary to complete the goal |
| action | String | The action to be taken as well as the necessary parameters for that action. Possible actions are tap(x,y), swipe(x,y), navigate({back, home, enter}), status({complete, impossible}) |
| goal | String | Text description of the task the agent is expected to perform in the current episode |
| action_history | List | List of previous actions taken at prior steps |
| xml | String | Path to the XML file |
| image | String | Path to the image file of the screen at the current step |
| image_history | List[String] | Paths to images at previous steps |
| complete | Boolean | whether a step is missing from the episode in the jsonl file |
| eval_category | String | Relevant during evaluation, describes whether the goal is one of the following: SEEN,NOVEL,FAMILIAR. |
| conversations | List[Dict] | The prompt provided to the model and the model's expected response, the action to be performed for the next step: |

Table 5: Description of fields used in the dataset.

## H  DATASET FORMAT

The format of the dataset is described in Table 5 and the action space we used is described in Table 6

| Action | parameters | examples |
|---|---|---|
| type | <text> | type('wooden toy') |
| tap | $(x, y)$ | tap(.23, .76) |
| swipe | $(x_1, y_1, x_2, y_2)$ | swipe(.43, .80, .51, .32) |
| navigate | back, home, enter | navigate(back) |
| status | complete, impossible | status(complete) |

Table 6: Action space used in our experiments. We convert all device control's datasets action space to a shared one for unified training.

## I  LICENSE OF ASSETS

Here is the license of pretrained models and datasets we use in the paper.

1. PerpcetionLM: We use `https://github.com/facebookresearch/perception_models`. The license is FAIR Noncommercial Research License.

2. Llama3: the license is Llama 3 community license available at `https://github.com/meta-llama/llama-models/blob/main/models/llama3_1/LICENSE` and `https://github.com/meta-llama/llama-models/blob/main/models/llama3_2/LICENSE`.

3. Llama4: the license is Llama 4 community license available at `https://github.com/meta-llama/llama-models/blob/main/models/llama4/LICENSE?ref=maginative.com`.

4. GPT4o. We use the API call via Azure. Term of use is available at `https://openai.com/policies/row-terms-of-use/`.

5. Android-in-the-wild: We use `https://github.com/google-research/google-research/tree/master/android_in_the_wild`. The license is Apache 2.0.

6. Android Control: We use `https://github.com/google-research/google-research/tree/master/android_control`. The license is Apache 2.0.

7. Cauldron: We use https://huggingface.co/datasets/HuggingFaceM4/the_cauldron. The license is CC-BY-4.0.

## J  PROMPTS

```
Android Agent Prompt

Assist an Android user by generating actions based on their conversational
input and the current screen image.

Available actions (pick one):
- tap(x, y): Tap at screen location (x, y). Example: tap(0.312, 0.589).
- swipe(x1, y1, x2, y2): Swipe from (x1, y1) to (x2, y2).  Example:
swipe(0.171, 0.350, 0.899, 0.357).
- type(text): Type text. Example: type(Hello).
- navigate(option): Navigate options: back, home, enter.  Example:
navigate(back).
- status(option): Status options: complete, impossible.  Example:
status(complete).

Please respond with a single action, with no additional text.

Goal: {goal}
Past actions: {past three actions}
What action should the user take next?
```

Prompt 1: Default prompt used for training and evaluating device control agents.

---

**Android Agent Prompt with CoT**

```
You are an agent operating an Android device and your task is to issue
ACTIONs to achieve a GOAL. It may require multiple steps of issuing ACTIONs
to achieve the GOAL. You are given the goal, the current STATE of the
Android device which is a screenshot, as well as HISTORY of the previous
STEP INFO. You need to generate the next STEP.

Each STEP X contains STEP INFO defined as follows:
(1) OBSERVATION - describe the screenshot
(2) THOUGHT - the thought process for what is the best ACTION in this step
to achieve the GOAL
(3) ACTION SHORT - a brief description of what the ACTION is. i.e. tap on
<element>
(4) EXPECTED UI CHANGE - after executing the ACTION, how do you expect the
UI to change
(5) ACTION - the precise interaction instruction to execute on the phone
as described below

ACTIONs are in the form:
tap(x, y) - tap on location (x, y) in normalized coordinates from 0 to 1
on the screen.
swipe(x1, y1, x2, y2) - swipe from location (x1, y1) to location (x2, y2)
in normalized coordinates from 0 to 1 on the screen.
navigate(home|back|enter) - press android soft keys (home, back, enter)
type(text) - type text on the screen.
status(complete|impossible) - end the trajectory because goal has been
completed or the goal is impossible.

The GOAL is "{goal}"

Last 3 HISTORY STEP INFOs: {last three steps model output}

Provide the next STEP INFO
```

Prompt 2: Prompt used to train and evaluate agents with CoT data.

```
Step Summarization Prompt

System Prompt:
You are an expert in summarizing the progress of an Android navigation
agent, whose role is to help a human user navigate the android phone to
complete a goal. The agent takes step-by-step phone actions like clicking,
swiping, typing, navigating home, navigating back, and ending.

User Prompt:
screen A: <image> screen B: <image> You are given two screenshots from an
Android navigation trajectory. The trajectory aims to achieve the goal of
"{goal}". The first screenshot (screen A) is before agent's action at a
particular step, and the second screenshot (screen B) is after the action.
The agent's action at this step is {action}. If the action involves tapping
or long press, the tap location is shown on screen A with a hollow green
circle. If the action involves swiping, the start and end points of the
swipe are shown on Screen A with a green arrow.

Here are the list of UI elements extracted from screen A and screen
B.
screen A: {simplified XML for screen A}
screen B: {simplified XML for screen B}
The list of UI elements may be incomplete and cannot accurately reflect
the changes in the screenshot image.You should focus on the screenshot
image if provided and only use the list of UI elements as complementary
information.

Summarize screen A, screen B, and the changes from A to B. Focus
on:
- What key elements (e.g., icons, texts, buttons, navigation bars, images,
forms and input field, pop-ups, switches and toggles etc) are there in
each screenshot and where are they (e.g., top-left, bottom-right etc).
- What did the agent do to get from A to B. If the action involves tapping,
infer which key element is tapped based on the action, hollow green circle
location and the context of the screenshot. If the action involves swiping,
infer the swipe direction based on the action, green arrow and the context
of the screenshot.
- What are the changes from A to B related to the goal. Pay attention to
subtle changes including altered or new key elements.

Be concise and comprehensive.  Only mention key elements which are
related to the goal and ignore the irrelevant ones. Your answer should be
in the following format, ensuring each section is no more than 100 words:
screen A: <summary of screen A>
screen B: <summary of screen B>
changes: <changes from A to B>
```

Prompt 3: Prompt for generating the step summary used in the automated success detection pipeline to be provided as input to the LLM judge.

**LLM Judge System Prompt**

```
System Prompt:
You are an expert in evaluating the performance of an Android navigation agent,
whose role is to help a human user navigate the android phone to complete a
goal. The agent takes step-by-step phone actions like clicking, swiping, typing,
navigating home, navigating back, and ending.

You are given the goal, the initial and final states of the android system.
For each action that the agent takes, you are also provided with a summary of the
transition from previous state (State A) to next state (State B) due to the action.

Your task is to evaluate whether the goal was achieved and stays achieved
during the episode. Please respond first with a reasoning of your evaluation based
on (i) the conditions required to fulfill the goal, and (ii) insights drawn from
all relevant state transitions concerning these conditions.  Do NOT repeat the
summary of the transition for each step in your reasoning. Then, respond with your
Yes/No judgment. Your answer should be in the following format:
Reason: <reasoning>
Judgment: <Yes/No>

Tips for deciding your evaluation:
- The ending action shows the agent's decision to stop the episode, but it doesn't
 necessarily indicate goal completion. You should deduce if the goal was achieved
 based on all previous state transitions and the agent's action.
- The agent should always issue an ending action in the final step to stop the
 episode. Failure to do so indicates goal not achieved.
- Consider the screen types of the initial or final states, such as:  home
 (workspace), list of settings, inside an app, status menu etc.
- There may be actions or states at any step that are irrelevant to the goal. Focus
 only on the relevant state transitions and reason agent's intent behind them,
 ignore the irrelevant steps.
- The goal is considered achieved as long as it is reached at any step and maintained
 thereafter, even if subsequent irrelevant actions divert the focus. However, if the
 goal is initially met but later reversed, it should be considered as not achieved.
- For information-seeking tasks, such as retrieving information from the Android
 phone, the intermediate or final state should either provide the requested
 information or state its unavailability.  Goal is achieved with information
 presented in any text format, including nested or secondary text.
- Sometimes the goal is achieved by default, not as a result of the agent's actions,
 such as when the objective is to activate a feature that is already enabled (like
 a toggle that's on or a checkbox that's checked), there's no need to interact with
 it.  If information at any state indicates that the goal is met and maintained
 thereafter, it should be considered goal achieved.
- If the goal is to enable, toggle, or turn on a feature, pay attention to the
 checkbox or toggle state.  Whether it is already enabled or the agent interacts
 with it to enable it, both are considered goal achieved.
- When the agent follows the necessary steps but is blocked by external factors,
 like trying to add an unavailable item to the cart or checking unavailable
 information, the effort is still considered as achieving the goal.

Be critical and conservative. Before saying yes, make sure that:
- The full goal must be achieved, not partial. For example, if the goal is to open
 a website, ensure it is opened, not just entering the URL in the search bar.
- For tasks that specify parameters for the goal, all the parameters must be
 correctly applied by the agent.
- For tasks that involve saving or creating, ensure that the agent has executed
 a save action or a comparable action. You will not receive confirmation that the
 save occurred, but you can assume it was successful if you observe that the agent
 clicked a save button or performed an equivalent action.
- If the goal involves opening a specific app, don't confuse it with similar apps,
 such as Google vs. Chrome.
- If the goal involves opening a specific setting, don't confuse it with similar
 settings, such as privacy vs. security.
```

Prompt 4: System prompt given to the LLM judge to evaluate success of a trajectory given a goal.

**LLM Judge User Prompt**

```
User Prompt:
Initial screenshot: <image> Final screenshot: <image> You are given two
screenshots from an Android navigation trajectory. The first screenshot
(Initial Screenshot) shows the initial state of the android system, and
the second screenshot (Final Screenshot) shows the final state.

The goal of the trajectory is "{goal}".

The agent took {total number of actions} actions.  Here is a history
of what the agent has done: {step-wise output from Step Summarization}.

Here are the list of UI elements extracted from the initial  and
final state of trajectory.
Initial State: {simplified XML for initial screenshot}
Final State: {simplified XML for final screenshot}
The list of UI elements may be incomplete and cannot accurately reflect the
initial and final state. You should focus on the screenshot if provided
and only use the list of UI elements as complementary information.
```

Prompt 5: Prompt given to the LLM judge to evaluate success of a trajectory given a goal.

**GPT4o Prompt**

System Prompt:
Imagine that you are imitating humans operating an Android device for a task step by step. At each stage, you can see the Android screen like humans by a screenshot and know the previous actions before the current step decided by yourself through recorded history. You need to decide on the first following action to take. You can tap on an element, long-press an element, swipe, input text, open an app, or use the keyboard enter, home, or back key. (For your understanding, they are like 'adb shell input tap', 'adb shell input swipe', 'adb shell input text', 'adb shell am start -n', and 'adb shell input keyevent'). One next step means one operation within these actions. Unlike humans, for typing (e.g., in text areas, text boxes), you should try directly typing the input or selecting the choice, bypassing the need for an initial click. You should not attempt to create accounts, log in or do the final submission. Terminate when you deem the task complete or if it requires potentially harmful actions.

User Prompt:
You are asked to complete the following task: {app_prefix} {goal}

Previous Actions:
{action_history}

The screenshot below shows the Android screen you see. Follow the following guidance to think step by step before outlining the next action step at the current stage:

(Current Screen Identification)
Firstly, think about what the current screen is.
{ui_raw}

(Previous Action Analysis)
Secondly, combined with the screenshot, analyze each step of the previous action history and their intention one by one. Particularly, pay more attention to the last step, which may be more related to what you should do now as the next step. Specifically, if the last action involved a INPUT TEXT, always evaluate whether it necessitates a confirmation step, because typically a single INPUT TEXT action does not make effect. (often, simply pressing 'Enter', assuming the default element involved in the last action, unless other clear elements are present for operation).

(Screenshot Details Analysis)
Closely examine the screenshot to check the status of every part of the screen to understand what you can operate with and what has been set or completed. You should closely examine the screenshot details to see what steps have been completed by previous actions even though you are given the textual previous actions. Because the textual history may not clearly and sufficiently record some effects of previous actions, you should closely evaluate the status of every part of the screen to understand what you have done.

(Next Action Based on Android screen and Analysis)
Then, based on your analysis, in conjunction with human phone operation habits and the logic of app design, decide on the following action. And clearly outline which element on the Android screen users will operate with as the first next target element, its detailed location, and the corresponding operation.

To be successful, it is important to follow the following rules:
1. You should only issue a valid action given the current observation.
2. You should only issue one action at a time
3. For handling the select dropdown elements on a screen, it's not necessary for you to provide completely
accurate options right now. The full list of options for these elements will be supplied later.

Prompt 6: Prompt for GPT4o baseline.

1296
1297
1298
1299
1300
1301
1302
1303
1304
1305
1306
1307
1308
1309
1310
1311
1312
1313
1314
1315
1316
1317
1318
1319
1320
1321
1322
1323
1324
1325
1326
1327
1328
1329
1330
1331
1332
1333
1334
1335
1336
1337
1338
1339
1340
1341
1342
1343
1344
1345
1346
1347
1348
1349

**QWen2.5VL Prompt**

```
System Prompt:
You are a helpful assistant.

# Tools

You may call one or more functions to assist with the user query.

You are provided with function signatures within <tools></tools> XML tags:
<tools>
{"type":   "function",  "function":   {"name_for_human":   "mobile_use",  "name":
"mobile_use", "description": "Use a touchscreen to interact with a mobile device,
and take screenshots.\n* This is an interface to a mobile device with touchscreen.
You can perform actions like clicking, typing, swiping, etc.\n* Some applications
may take time to start or process actions, so you may need to wait and take
successive screenshots to see the results of your actions.\n* The screen's
resolution is 1092x2408.\n* Make sure to click any buttons, links, icons, etc with
the cursor tip in the center of the element. Don't click boxes on their edges unless
asked.", "parameters": {"properties": {"action": {"description": "The action to
perform. The available actions are:\n* 'click': Click the point on the screen with
coordinate (x, y).\n* 'long_press': Press the point on the screen with coordinate (x,
y) for specified seconds.\n* 'swipe': Swipe from the starting point with coordinate
(x, y) to the end point with coordinates2 (x2, y2).\n* 'type': Input the specified
text into the activated input box.\n* 'system_button': Press the system button.\n*
'terminate':  Terminate  the  current  task  and  report  its  completion  status.",
"enum": ["click", "long_press", "swipe", "type", "system_button", "terminate"],
"type":  "string"}, "coordinate": {"description":  "(x, y): The x (pixels from
the left edge) and y (pixels from the top edge) coordinates to move the mouse
to.  Required only by 'action=click', 'action=long_press', and 'action=swipe'.",
"type":  "array"}, "coordinate2": {"description":  "(x, y): The x (pixels from
the left edge) and y (pixels from the top edge) coordinates to move the mouse
to. Required only by 'action=swipe'.", "type": "array"}, "text": {"description":
"Required only by 'action=type'.", "type":  "string"}, "time": {"description":
"The seconds to wait. Required only by 'action=long_press'.", "type": "number"},
"button": {"description": "Back means returning to the previous interface, and Home
means returning to the desktop. Required only by 'action=system_button'", "enum":
["Back", "Home"], "type": "string"}, "status": {"description": "The status of the
task. Required only by 'action=terminate'.", "type": "string", "enum": ["success",
"failure"]}}, "required": ["action"], "type": "object"}, "args_format": "Format
the arguments as a JSON object."}}
</tools>

For each function call, return a json object with function name and arguments within
<tool_call></tool_call> XML tags:
<tool_call> {"name": <function-name>, "arguments": <args-json-object>} </tool_call>
User Prompt:
The user query: {app_prefix} {goal}
Task progress (You have done the following operation on the current device):
{action_history}
```

Prompt 7: Prompt for QWen2.5VL baseline

## Chain-of-Thought Generation Prompt

```
System Prompt:
You are an expert in summarizing the progress of an Android navigation agent, whose
role is to help a human user navigate the android phone to complete a goal. The
agent takes step-by-step phone actions like clicking, swiping, typing, navigating
home, navigating back, and ending.

User Prompt:
screen A: <image> screen B: <image> You are given two screenshots from an Android
navigation trajectory. The trajectory aims to achieve the goal of "{goal}". The
first screenshot (screen A) is before agent's action at a particular step, and the
second screenshot (screen B) is after the action. The agent's action at this step
is {action}. If the action involves tapping or long press, the tap location is
shown on screen A with a hollow green circle. If the action involves swiping, the
start and end points of the swipe are shown on Screen A with a green arrow.

Here are the list of UI elements extracted from screen A and screen B.
screen A: {simplified XML for screen A}
screen B: {simplified XML for screen B}
The list of UI elements may be incomplete and cannot accurately reflect the changes
in the screenshot image.You should focus on the screenshot image if provided and
only use the list of UI elements as complementary information.

First summarize screen A, screen B. Focus on:
- What key elements (e.g., icons, texts, buttons, navigation bars, images, forms
and input field, pop-ups, switches and toggles etc) are there in each screenshot
and where are they (e.g., top-left, bottom-right etc).
- Do NOT mention the terms "screen A" or "screen B" in your response. Instead,
refer to them as the screenshot.

Next describe the agent's action.  Make it short, in one sentence.  Focus
on:
- What did the agent do to get from A to B. If the action involves tapping, infer
which key element is tapped based on the action, hollow green circle location and
the context of the screenshot. If the action involves swiping, infer the swipe
direction based on the action, green arrow and the context of the screenshot.
- Do NOT mention the terms "hollow green circle" or "green arrow" in your response.

Then summarize the changes from A to B. Focus on:
- What are the changes from A to B related to the goal. Pay attention to subtle
changes including altered or new key elements.
- Do NOT mention the terms "screen A" or "screen B" in your response.  Refer to
screen A as the current screenshot and screen B as the next screenshot.

Finally provide a reasoning on why the agent took this action. Focus on:
- What are the reasons for the agent to take this action? Why does this action help
the agent to achieve the goal?

Be concise and comprehensive.  Only mention key elements which are related
to the goal and ignore the irrelevant ones.
Your answer should be in the following format, ensuring each section is no more
than 100 words:
screen A: <summary of screen A>
screen B: <summary of screen B>
action description: <action description>
changes: <changes from A to B>
reason: <why took this action>
```

Prompt 8: Prompt for generating the Chain-of-Thought data.

## K  DATA VERIFICATION

We ensure the quality of our data by employing human reviewers to all trajectories. We trained over 100 contractors organized into 2 cohorts to perform the validation task. We created a training curriculum as follows: Annotators-in-training practiced verification tasks on alternative datasets such as AITW. The team annotated an initial golden dataset with DigiData trajectories. Annotators were then given a qualification test based on the golden dataset. This test includes strict guidelines in the form of a grading rubric to ensure the annotators have fully learned the task. In addition to verifying the trajectory level correctness, the annotators also are tasked with verifying the validity of the goals generated from our goal generation pipeline. We have each trajectory reviewed by two human verifiers and consider a trajectory as achieving a goal if both agree.

| Error Name | Error Type | Description | Point Deduction Production | Point Deduction Training |
|---|---|---|---|---|
| Incorrect or missing goal parameters | Critical | No selection is made for goal parameters or an incorrect selection is made | 11 | 16 |
| Incorrect or missing trajectory question | Critical | No selection is made for if the trajectory achieves the goal or an incorrect selection is made | 11 | 16 |
| Incorrect or missing cause of failure | Critical | No selection is made for cause of failure when required or an incorrect selection is made | 11 | 16 |
| Incorrect or missing response for does the goal make sense | Critical | No selection is made for "does the goal make sense" or an incorrect selection is made | 11 | 16 |
| Incorrect or missing step accuracy question | Non-Critical | No selection is made for step accuracy or an incorrect selection is made | 5 | 8 |
| Step(s) missing text box descriptions or not following format | Non-Critical | Text box description is missing where it is required or the description does not meet quality standards | 5 | 8 |
| Step(s) incorrect or missing label accuracy question | Non-Critical | There is no selection made for label accuracy or an incorrect selection is made | 5 | 8 |
| Incorrect or missing critical to achieve this goal question | Non-Critical | No selection is made for if the step is critical or an incorrect selection is made | 5 | 8 |
| Incorrect or missing response for is there a step missing before this | Non-Critical | No selection is made for "is there a step missing before this" or incorrect selection is made | 5 | 8 |
| Spelling errors or sentence structure | Non-critical | Misspelled words or sentence structure make it difficult to understand explanations | 2 | 4 |
| Extra questions answered | Non-critical | There are more questions answered than there are steps | 2 | 4 |

Table 7: Error Table and Point Deductions

## L  GOAL GENERATION PROTOCOL

We consider the quality of generated goals to be as crucial as the quality of collected trajectories, since the goal distribution directly shapes the overall dataset distribution. To ensure a high-quality goal set that offers broad coverage of app features, approximates natural user behaviors, and maintains strong diversity, we collaborate closely with two research assistants who adhere to a rigorous protocol refined through multiple rounds of feedback.

**To ensure comprehensive coverage**, research assistants are provided with Android devices containing the target apps. They systematically explore all pages and interactable elements within each app, creating goals associated with these pages and elements.

**To approximate natural user behaviors**, research assistants are instructed to devise goals that mirror typical user intentions, rather than relying on overly mechanical templates such as repetitive `click on X` phrasing. Instead, research assistants are encouraged to express goals in natural language, as they would in everyday scenarios. We also promote the use of diverse verbs and varied grammatical structures to prevent the goals from appearing uniform or formulaic.

**To promote diversity**, we introduce the concepts of goal templating and parameterization. Research assistants first generate unique template formats with placeholder parameters (e.g., `Save the article named <article_name> to my profile`). These templates are then instantiated with specific arguments to create concrete goals. In this framework, templates are considered functionally distinct (e.g., `Search for an article named <article_name>` versus `Change language to <language_name>`).

**To further ensure goal quality**, we implement a multi-stage review and filtering process using several sources of feedback. First, researchers examine the generated goals to check their quality and provide feedback, allowing research assistants to revise or remove problematic goals. Second, data annotators flag goals that are unclear or not feasible for data collection, and these are excluded.

Third, during data verification, judges assess whether each goal is meaningful and achievable; any goals found invalid are removed from the final dataset. Through these combined review stages, we continually refine the goal set to maintain high standards.

## M  RUNBOOKS

---

### Eval Runbook

This eval requires you to run an AI model at each step to automatically operate an Android phone. After completing a trajectory, you will then be asked to upload it after filling out a quick survey.

Ensure that all of the following apps are installed:

Airbnb, Alibaba.com, Aliexpress, Amazon Shopping, Calculator, Calendar, Camera, Cars.com, Clock, Doordash, Easybudget, Ebay, Expedia, Gallery (Google), Gmail, Google, Google Chrome, Google Maps, Google Messages, Google Photos, Google Play Store, Google Translate, Inshot, Live Transcribe & Notification, Mcdonald's, Pdf Reader, Phone, Settings, Shein, Temu, The Weather Channel - Radar, Walmart, Waze, Wikipedia, Wish, and Zoom

For each Atomic Digital Goal (ADG), make sure you set up your app before evaluating the ADG. This means completing any required pre-work and opening the app to its homepage. Once that is done, follow these instructions:

1. Select the model you were instructed to use.

2. Enter the ADG into the goal field.

3. Press actuate to generate an action.

4. Assess if the generated action is **DANGEROUS** (see Dangerous Action Guidelines). If it falls into one of those categories, do **NOT** confirm it. Instead, upload the trajectory as is and complete the survey saying the goal was not achieved and that the reason was for a dangerous action.

5. If you don't consider the action to be dangerous, then press confirm. The action should then be executed on your phone.

6. Wait for the screen to settle (animations, loading/progress indicators, etc.). Then press actuate again and repeat the process from Step 3. Keep doing this until one of the following happens:

    a. The model returns a complete action.

    b. The model starts looping, meaning it produces the exact same action or set of actions three or more times in a row.

    c. The model reaches the maximum number of steps (25) allowed.

    d. The model suggests a dangerous action.

    e. The model suggests inaccurate actions that will disallow the goal from being achieved.

7. Once one of these conditions is met, fill out the survey and upload your trajectory. The survey will ask you if the goal was successfully achieved. If it wasn't, then select the best reason for the model failing:

    a. **Missing Final Step**: The model produced the complete action one step too soon. This means the model believes the goal is achieved, but the UI is actually in a state where you would need exactly one more action to fully complete the goal.

    b. **Impossible**: This goal is not possible to be completed, even if you were to attempt this goal manually.

    c. **Overshot**: The model at one point did achieve the goal. However, it kept on suggesting other actions after it had completed the goal.

    d. **Dangerous**: The suggested action falls into the guidelines for dangerous actions.

    e. **Looping**: The model started looping through the exact same action or set of actions three or more times in a row.

    f. **Max Steps**: The maximum number of steps allowed, 25, was reached.

    g. **Inaccurate**: The model suggested actions that would cause it to fail in achieving the goal.

Runbook 1: Eval runbook given to human annotators

---

**Dangerous Action Guidelines**

Please note that dangerous actions are actions in the sequence that can actually 'write' to the app. Preceding actions in the sequence are not considered dangerous actions (e.g. for the goal 'Leave a 5-star review in a restaurant on Yelp':

1. Open Yelp
2. Open restaurant page
3. Choose 5 stars
4. Write the comment
5. Press post review button

Only the 5th action is considered dangerous because the first 4 steps didn't change the state of the app until the 5th one got issued. The following are the various categorizations of why an action would be considered dangerous:

**Social Interactions and Online Presence**
- Posts comments or messages
- Shares posts or updates
- Sends direct messages
- Blocks or reports users
- Accepts/Declines friend requests
- Joins online groups or communities

**Financial Transactions**
- Presses purchase button
- Confirms payment method
- Enters credit card information
- Authorizes transactions or sends money to others

**Account Management and Security**
- Changes password
- Links account to other services/accounts
- Deletes account

---

Runbook 2: Guidelines given to human annotators on what should be classified as a dangerous action

---

**Data Collection Runbook**

This study aims to understand how users interact with their mobile devices. During the study, you will be directed to an external webpage where you will interact with an Android phone. Your goal is to perform the following action as directly as possible: <atomic_digital_goal> following these steps:

1. Click on the 'Start Study' button to begin the data collection.

2. Read instructions and familiarize yourself with the data collection web tool before starting the task.

3. Find the required App, which is included in the task instructions and start completing the task.

4. When typing: use the virtual keyboard and tap on each key individually. Wait for text to appear before moving on to the next key

5. When completed the task, click on 'Submitted Completed Goal' to submit the result.

6. There are 3 types of successful submissions and 1 type of failure submission:

   a. Failure submission: if the task is not possible to complete, toggle the switch to 'FAILURE', enter the reason then submit.

   b. Successful type 1 – Information retrieval with direct return value: if the task requires to return a specific piece of information, enter the retrieved information into the result text box and submit.

   c. Successful type 2 – Information retrieval with long output in screen: if the information required is too long to enter as text output, use mouse to drag and select the area of information needed, then submit.

   d. Successful type 3 – browse or side effect: if the task only asks to complete an action without asking for information, skip text entry and click 'RECORD NO OUTPUT' to submit.

7. Do **NOT** provide any of your personal information or sign in, view, access any personal accounts at any time during the whole process of the data collection. We prepared the necessary test information in the task instructions that will be sufficient to help you complete the task.

Runbook 3: Data collection runbook given to human annotators

