# OpenReview forum: "DigiData: Training and Evaluating General-Purpose Mobile Control Agents"
_ICLR.cc/2026/Conference — Submitted to ICLR 2026_

### Official Review · Reviewer_Fk85 · 2025-10-30

**Soundness:** 3
**Presentation:** 3
**Contribution:** 3
**Rating:** 4
**Confidence:** 3

**Summary:**

DigiData is presented as a large-scale, high-quality, diverse, multi-modal dataset engineered for training general-purpose mobile control agents. DigiData employs a multi-phase collection protocol involving goal curation through comprehensive app feature exploration, human demonstrations, and trajectory verification to ensure high data quality and complexity. Digidata comprises of 152k trajectories and 8k unique goals across 26 Android apps. DigiData-Bench is also provided, a benchmark that introduces dynamic evaluation protocols, supported by human-assisted methods, demonstrating that task success rate is required to reliably assess agent capabilities, as the traditional step-accuracy metric is often insufficient and unreliable for performance ranking

**Strengths:**

* The contribution is well motivated: constructing large-scale datasets is essential in advancing and developing mobile device control agents.
* The multi-phase data collection pipeline with humans incorporated in each phase ensures that high-quality data is collected.
* The dataset is comprehensively constructed for each app, represented by the number of tasks per app.

**Weaknesses:**

* In Table 2, there is limited comparison between existing agents and models fine-tuned using DigiData, which limits understanding of the amount of improvement in mobile control capability DigiData provides. As this is an important aspect of this work, I suggest evaluating more mobile agents on DigiData-Bench. It would also help to include PLM as a baseline, as it will help understand the magnitude of improvement from the base model via fine-tuning on DigiData.
* The number of tasks per app appears quite large. While the proposed similarity metric suggests that the tasks remain diverse, it is difficult to assess the actual degree of diversity without reviewing the task lists themselves. Conceptually, there is a natural limit to the number of genuinely distinct tasks an app, especially simple ones like Google Calendar or Google Clock, can support. This raises concerns that some tasks may be variations of the same or very similar actions. Providing examples or additional evidence of task distinctiveness would strengthen the validity of this claim.

**Questions:**

* Are there any evaluation results for DigiData-Bench-Auto? This benchmark seems likely to be more generally used by other researchers. Providing insights related to the auto benchmark would be useful.

I am willing to raise my score given that my questions or weaknesses are addressed.

---

> ### Author Response · Authors · 2025-11-27
> **Response to Reviewer Fk85**
>
> # 1. Comparison with Existing Agents and Inclusion of PLM Baseline
>
> > **W1**: In Table 2, there is limited comparison between existing agents and models fine-tuned using DigiData, which limits understanding of the amount of improvement in mobile control capability DigiData provides. As this is an important aspect of this work, I suggest evaluating more mobile agents on DigiData-Bench. It would also help to include PLM as a baseline, as it will help understand the magnitude of improvement from the base model via fine-tuning on DigiData.
>
> We agree that a broader comparison with existing agents would further clarify the impact of DigiData. While our current evaluation focuses on the most relevant and accessible baselines, we are actively working to expand this set. Regarding the PLM baseline, we agreed that including it will help understand the magnitude of improvement from the base model via fine-tuning on DigiData. We are currently running experiments and will update the paper as soon as we obtain the full results. As a preliminary check, we observed that the step accuracy of the PLM 8B model (without fine-tuning) on DigiData-Bench is 1%. In comparison, the fine-tuned PLM 8B achieves 70.7% step accuracy, and the fine-tuned PLM 8B with COT reaches 72.8%. This indicates that the model initially has nearly zero capability in mobile agent navigation, but gains significant improvement after fine-tuning on DigiData.
>
> ---
>
> # 2. Task Diversity and Distinctiveness
>
> > **W2**: The number of tasks per app appears quite large. While the proposed similarity metric suggests that the tasks remain diverse, it is difficult to assess the actual degree of diversity without reviewing the task lists themselves. Conceptually, there is a natural limit to the number of genuinely distinct tasks an app, especially simple ones like Google Calendar or Google Clock, can support. This raises concerns that some tasks may be variations of the same or very similar actions. Providing examples or additional evidence of task distinctiveness would strengthen the validity of this claim.
>
> We appreciate the request for more concrete evidence of task diversity. Qualitatively, we can provide examples that highlight the granularity and distinctiveness of our task definitions. We have attached a Google Colab notebook in the footnote of the first page, which allows for direct examination of the dataset and task lists. This should help clarify how we enumerate all possible features/tasks for each app and demonstrate the distinctiveness of each task. Additionally, we will include a table of representative task examples per app in the appendix to further illustrate this diversity.
>
> ---
>
> # 3. Evaluation Results for DigiData-Bench-Auto
>
> > **Q1**: Are there any evaluation results for DigiData-Bench-Auto? This benchmark seems likely to be more generally used by other researchers. Providing insights related to the auto benchmark would be useful.
>
> Thank you for highlighting the importance of DigiData-Bench-Auto. We agree that this benchmark will be of broad interest to the community, and our long-term goal is to encourage community engagement and maintain a leaderboard of all relevant work. We will include the evaluation results for DigiData-Bench-Auto in the updated version of the paper.
>
> ---
>
> # Summary
>
> We thank the reviewer for their constructive feedback and willingness to reconsider their score. We are committed to addressing all points raised:
>
> - Expanding baseline comparisons (including vanilla PLM results on DigiData-Bench)
> - Providing qualitative evidence of task diversity
> - Reporting results for DigiData-Bench-Auto
>
> We believe these additions will further strengthen the submission and clarify the contributions of DigiData to the community.

---

### Official Review · Reviewer_PvGF · 2025-10-31

**Soundness:** 2
**Presentation:** 2
**Contribution:** 2
**Rating:** 4
**Confidence:** 4

**Summary:**

This paper introduces DigiData, a large-scale mobile GUI dataset comprising 8,275 unique goals across 26 Android applications. The dataset was constructed by skilled annotators following a goal generation protocol and is characterized by its high complexity (average of 9.2 steps/goal), multi-modal nature (including screenshots, UI trees, and Chain-of-Thought annotations), and high quality (almost 100% trajectory verification pass rate). Furthermore, the authors introduce the DigiData-Bench, an online evaluation benchmark that supports task assessment by either LLMs or human evaluators based on an evaluation protocol. The authors use this dataset to fine-tune the Perception Language Model. Experiments show that an 8B parameter model trained with Chain-of-Thought (CoT) data outperforms existing baseline models.

**Strengths:**

1.	The paper's appendix provides exhaustive details on data construction and in-depth data analysis.
2.	By leveraging skilled annotators who systematically explore application functionalities based on a goal generation protocol, the generated goals cover advanced features of the apps, enhancing the dataset's depth and utility.
3.	DigiData is currently the second-largest mobile control dataset. It boasts high quality (100% trajectory verification pass rate) and provides rich information, including screenshots, UI trees, and LLM-generated Chain-of-Thought data, offering a valuable resource for training GUI agents.
4.	The paper conducts an detailed evaluation of using LLM judges for GUI tasks, providing valuable empirical evidence.

**Weaknesses:**

1. The experimental evaluation covers limited datasets and benchmarks, omitting important comparisons such as the GUIOdyssey [1] dataset, which features longer interaction sequences (average 15.3 steps) than the proposed DigiData.
2. The experimental design lacks consistent evaluation conditions and ablation studies, making it unclear whether DigiData independently enhances generalization. Current results merely show good performance on AitW and DigiData-Bench, without achieving SOTA on AndroidControl. Demonstrating cross-benchmark improvements would strengthen the argument.
3. Evaluation on DigiData-Bench involves only two zero-shot models (Qwen2.5VL and GPT-4o) and omits comparisons with open-source GUI agents. This limits the validation of task difficulty and dataset utility. Including models like OdysseyAgent [1] or GUI-Owl [3] would provide a more comprehensive assessment.
4. According to [2], GUI step accuracies can be approximated as i.i.d., implying that task success ≈ (step accuracy)^steps. However, in Table 2, GPT-4o’s step accuracy (40.0%) and task success rate (27.8%) deviate substantially from expected ratios and from reports such as OdysseyAgent [2] (3.22% success at 59.53% step accuracy). The reported success rates thus appear inconsistent and require clarification.
5. Although Figure 5 shows some correlation between LLM-as-Judge and human evaluations, the noticeable gap suggests that LLM-as-Judge cannot yet provide fully reliable absolute performance estimates, warranting further validation.

[1] Lu, Quanfeng, et al. ”GUIOdyssey: A Comprehensive Dataset for Cross-App GUI Navigation on Mobile Devices.”

[2] Li W, Bishop W E, Li A, et al. On the effects of data scale on ui control agents[J]. Advances in Neural Information Processing Systems, 2024, 37: 92130-92154.

[3] GUI-Owl，Ye J, Zhang X, Xu H, et al. Mobile-agent-v3: Fundamental agents for gui automation[J]. arXiv preprint arXiv:2508.15144, 2025.

**Questions:**

1. DigiData includes only 26 Android applications, which is significantly fewer than datasets such as AitW, AndroidControl, and GUIOdyssey. How does this limited app coverage affect the dataset’s representativeness and generalizability? The authors are encouraged to justify why this scale is sufficient for training general-purpose GUI agents.
2. The paper does not clearly explain how the benchmark goals in DigiData-Bench were selected. Please elaborate on the filtering criteria, the balance of goal distributions, and the measures taken to avoid tasks that are either too simple or excessively difficult.
3. The validity of DigiData-Bench depends heavily on its evaluation protocol. More details are needed regarding how this protocol was constructed and how its accuracy and reliability are ensured.
4. Both human-assisted evaluation and LLM-as-Judge incur significant costs, with human evaluation being particularly time-intensive. The paper does not report the deployment cost of DigiData-Bench (e.g., time overhead or computational resources). Please provide information on the evaluation efficiency and overall cost.

---

> ### Author Response · Authors · 2025-11-27
> **Response to Reviewer PvGF [1/3]**
>
> # 1. Experimental Evaluation and Dataset Comparisons
>
> > **W1**: The experimental evaluation covers limited datasets and benchmarks, omitting important comparisons such as the GUIOdyssey [1] dataset, which features longer interaction sequences (average 15.3 steps) than the proposed DigiData.
>
> We acknowledge the value of including GUIOdyssey in our comparative analysis. We are actively working to provide numbers on GUIOdyssey, and will update the paper as soon as these results are available.
>
> ---
>
> # 2. Experimental Design, Ablation Studies, and Generalization
>
> > **W2**: The experimental design lacks consistent evaluation conditions and ablation studies, making it unclear whether DigiData independently enhances generalization. Current results merely show good performance on AitW and DigiData-Bench, without achieving SOTA on AndroidControl. Demonstrating cross-benchmark improvements would strengthen the argument.
>
> Our primary goal with DigiData is to provide a high-quality, diverse dataset and benchmark for the community, rather than to claim SOTA across all existing benchmarks. Achieving SOTA on every benchmark is not always meaningful for a dataset/benchmark paper, especially given the risk of overfitting to specific datasets.
>
> Notably, AndroidControl evaluates models on step accuracy, which can be problematic for assessing true generalization. Our experiments show that training solely on one dataset often leads to overfitting, resulting in high performance on that dataset but poor transfer to others. We believe generalization should be measured by reasonable performance across multiple benchmarks, not necessarily SOTA on all.
>
> ---
>
> # 3. Comparisons with Open-Source GUI Agents
>
> > **W3**:  Evaluation on DigiData-Bench involves only two zero-shot models (Qwen2.5VL and GPT-4o) and omits comparisons with open-source GUI agents. This limits the validation of task difficulty and dataset utility. Including models like OdysseyAgent [1] or GUI-Owl [3] would provide a more comprehensive assessment.
>
> We are working to include results for open-source agents such as OdysseyAgent [1] and GUI-Owl [3]. These comparisons will provide a more comprehensive assessment of DigiData-Bench’s difficulty and utility. We will update the paper with these results as soon as they are available.

---

> ### Author Response · Authors · 2025-11-27
> **Response to Reviewer PvGF [2/3]**
>
> # 4. Step Accuracy vs. Task Success Rate
> > **W4**: According to [2], GUI step accuracies can be approximated as i.i.d., implying that task success ≈ (step accuracy)^steps. However, in Table 2, GPT-4o’s step accuracy (40.0%) and task success rate (27.8%) deviate substantially from expected ratios and from reports such as OdysseyAgent [2] (3.22% success at 59.53% step accuracy). The reported success rates thus appear inconsistent and require clarification.
>
> Thank you for raising this important point. We would like to clarify in detail, as this is an important point we are raising in this paper.
>
> As noted in AndroidControl paper, the assumption that GUI step accuracies are i.i.d. is a simplifying assumption, arguably even an over-simplification, that does not hold in practice when evaluating task success in an online setting. Specifically, rather than evaluating an agent against a static, ground-truth trajectory (where every step must match the annotation for the task to be considered successful), our evaluation places the agent in a live, dynamic environment. Here, the agent is considered successful if it ultimately reaches the goal state and terminates, regardless of the specific route taken, which may differ significantly from the annotated ground-truth trajectory.
>
> A step marked as a failure in offline step evaluation may not constitute a failure in dynamic, online evaluation. This can occur in several scenarios:
>
> - The agent is exploring the app to gain necessary knowledge.
> - The agent is following an alternative valid route, as multiple paths may lead to success.
> - The agent makes a mistake but is able to recover in subsequent steps and still complete the task successfully.
>
> In fact, this is precisely the point we emphasize in Section 5.2, "Results - Online Evaluations Matter": step accuracy is not a reliable measurement of task success for the reasons outlined above. We also provide a concrete example in Figure 9 of the Appendix, where three distinct navigation modes are all valid for achieving the same goal. In such a case, if we pick one valid route as ground truth, but the agent is only capable of achieving the goal via another valid route, it will pass the online evaluation but might fail several offline step accuracy evaluations, which contradicts the i.i.d. assumption.
>
> This explains the substantial deviation from the expected ratio. To further clarify the difference with OdysseyAgent: OdysseyAgent’s success rate is calculated such that a task is considered successful only if all actions are correct, i.e., it requires every step to match the ground-truth annotation. In contrast, our evaluation measures success in an online, dynamic environment, where the i.i.d. assumption for step accuracy does not apply, and success is determined by the agent’s ability to complete the task in practice.
>
> ---
>
> # 5. LLM-as-Judge vs. Human Evaluation
> > **W5**: Although Figure 5 shows some correlation between LLM-as-Judge and human evaluations, the noticeable gap suggests that LLM-as-Judge cannot yet provide fully reliable absolute performance estimates, warranting further validation.
>
> We agree that LLM-as-Judge is not a perfect substitute for human evaluation, and further validation is needed. Our results show promising correlation, but absolute reliability remains an open question. We will emphasize this limitation and discuss future directions for improving LLM-based evaluation.
>
> ---
>
> # 6. App Coverage, Representativeness, and Generalizability
> > **Q1**:  DigiData includes only 26 Android applications, which is significantly fewer than datasets such as AitW, AndroidControl, and GUIOdyssey. How does this limited app coverage affect the dataset’s representativeness and generalizability? The authors are encouraged to justify why this scale is sufficient for training general-purpose GUI agents.
>
> We selected apps in a round-robin manner based on popularity and representativeness across diverse categories (e.g., Productivity, E-commerce, Finance, Connectivity, Chain-restaurant, Learning, Travel, Communication, System actions, News & Magazine, Tool, File viewer/player). This ensures coverage of a broad spectrum of real-world user interactions, despite the smaller app set.
>
> Unlike Android-in-the-wild or AndroidControl, which focus on breadth, DigiData emphasizes depth within each app, with an average of 318 goals per app. This approach ensures comprehensive coverage of app behaviors. Expanding to more apps is a future goal, and we agree this will further enhance representativeness.
>
> Regarding generalizability, we note that out-of-domain (OOD) generalization remains a challenge for all mobile agent datasets. For example, AitW observed significant OOD drops (see Table 4 of AitW), highlighting the difficulty of generalizing to unseen apps. We hope DigiData serves as a foundational step, and we encourage future work on exploration-based approaches (e.g., reinforcement learning) to address this challenge.

---

> ### Author Response · Authors · 2025-11-27
> **Response to Reviewer PvGF [3/3]**
>
> # 7. Benchmark Goal Selection and Evaluation Protocol
> > **Q2**: The paper does not clearly explain how the benchmark goals in DigiData-Bench were selected. Please elaborate on the filtering criteria, the balance of goal distributions, and the measures taken to avoid tasks that are either too simple or excessively difficult.
>
> We appreciate this feedback and will expand our description of the benchmark goal selection process. The goals were selected through a study involving 11 participants, with the following details:
>
> - **Diversity**: A total of 309 goals were generated from 37 apps, which are representative of 8 categories: Comms, Food, Management, Media, Navigation, Restricted Search, Shopping, and Misfit. The goals were contributed by 11 individuals who had no prior connection to the training dataset collection.
> - **Difficulty Level**: Goals are categorized based on execution intuitiveness, participants were requested to generate tasks of different difficulty levels:
>     - Intuitive goals: Easy to carry out by the average person without prior knowledge of the app’s inner workings; each step is straightforward and intuitive.
>     - Non-intuitive goals: Execution is not immediately obvious and may require searching or exploration within the app. The average person may need to browse the web to figure out how to accomplish these tasks.
> - **Splits**: Apps are also categorized based on their connection to the training dataset: seen, familiar, and novel.
> - **Post-processing**: Researchers reviewed and manually curated the initial set to ensure diversity and remove duplication. A professional data scientist was engaged in this process to ensure validity, diversity, and balance.
>
> As a result:
>
> - **Filtering Criteria**: The initial test set was generated under controlled conditions with clear guidelines, using an unbiased and sufficiently large group of participants. Subsequent filtering and curation involved direct researcher involvement and data scientist engagement to remove invalid, duplicate, overly simple, or excessively complex tasks.
> - **Balance of Goal Distribution**: Apps are balanced across 8 categories and 3 splits. Goals are distributed evenly across apps, with difficulty balanced between intuitive/easy (66%) and non-intuitive/hard (34%) tasks.
> - **Measures to Avoid Tasks That Are Too Simple or Too Difficult**: Detailed instructions and examples were provided during test set generation, followed by manual and scientific post-processing/curation to ensure appropriate task complexity.
>
> ---
>
> # 8. DigiData-Bench evaluation protocol details, accuracy and reliability
> > **Q3**: The validity of DigiData-Bench depends heavily on its evaluation protocol. More details are needed regarding how this protocol was constructed and how its accuracy and reliability are ensured.
>
> The runbook used by human evaluators is included in Appendix L.Runbooks – "Runbook 1: Eval runbook given to human annotators" for full transparency regarding our evaluation protocol.
>
> Additionally, the following measures are taken to ensure the accuracy and reliability of the evaluation process:
> - **Comprehensive Training**: All human evaluators are fully trained in advance, following the same standards and protocols.
> - **Consistent Task Assignment**: Evaluators are assigned consistent tasks across runs and have previously evaluated more than 10 other models on the same test set before evaluating the models in this paper.
> - **Direct Communication and Oversight**: The authors maintain direct communication with human evaluators to observe, monitor, identify, and resolve any issues before running evaluations for the models presented in this work.
> - **Author Review**: The authors review both the evaluation process and the results to ensure consistency, accuracy, and reliability.
>
> ---
>
> # 9. Evaluation Efficiency and Cost
> > **Q4**: Both human-assisted evaluation and LLM-as-Judge incur significant costs, with human evaluation being particularly time-intensive. The paper does not report the deployment cost of DigiData-Bench (e.g., time overhead or computational resources). Please provide information on the evaluation efficiency and overall cost.
>
> We agree that evaluation efficiency and cost are important considerations. Human evaluation is indeed time-intensive, while LLM-as-Judge offers a scalable alternative.
>
> **Human Evaluation Cost:**
>
> Evaluating one model on the full test set (309 goals) requires approximately 50–60 human hours (involving 5 human evaluators over 2 working days).
>
> **LLM-as-Judge Evaluation Cost:**
>
> The cost per trajectory is proportional to the number of steps (N) in the trajectory, as full verification requires (N + 1) LLM judge model API calls. Therefore, the total cost is calculated as: 309 × (average_trajectory_length + 1) × [model call cost]
>
> ---
>
> We appreciate the reviewer's feedback and hope these clarifications address your concerns.

---

### Official Review · Reviewer_FLBY · 2025-10-31

**Soundness:** 3
**Presentation:** 2
**Contribution:** 3
**Rating:** 4
**Confidence:** 5

**Summary:**

This paper presents DigiData, a large-scale, high-quality multimodal dataset (152k trajectories, 8k+ goals) for training general-purpose mobile control agents. It is built via a structured three-phase pipeline: goal curation, human demonstrations, and hybrid LLM–human verification, to ensure depth and diversity. The authors also propose DigiData-Bench, a dynamic benchmark featuring human- and AI-assisted evaluation protocols that address limitations of step-accuracy metrics. Experiments show DigiData-trained agents outperform existing baselines, and LLM judges correlate strongly with human assessments.

**Strengths:**

DigiData provides a well-structured and reproducible dataset with higher goal complexity and diversity than prior work. Experimental results are robust and clearly demonstrate performance gains and scaling effects. The dataset and benchmark together contribute to a valuable foundation for research in mobile control.

**Weaknesses:**

While DigiData is carefully engineered, it shows limited methodological novelty. Its contributions mainly concern dataset scale and organization rather than new paradigms. The motivation for creating DigiData is not clearly articulated, as existing datasets like AitW and AndroidControl already support complex mobile control research with similar design goals. The introduction of LLM judges is also incremental rather than innovative, and the paper does not situate its evaluation method within the broader literature on LLM-based judging in mobile control tasks.

**Questions:**

1.	The paper claims DigiData’s trajectories achieve 94.6% goal success (before filtering) compared to AitW’s 84%, but both were verified under the authors’ own system. Could this introduce evaluation bias or unfairness? How do the authors ensure comparability, and could a case study or qualitative inspection clarify what “quality difference” actually looks like?
2.	What concrete gaps or shortcomings in AitW and AndroidControl specifically motivated the creation of DigiData?
3.	Since the use of LLM-as-a-judge is not novel, how does this implementation differ from or improve upon prior research on automated trajectory evaluation?
4.	Can the authors benchmark their LLM judging pipeline against other existing studies applying LLMs for mobile control assessment?

---

> ### Author Response · Authors · 2025-11-27
> **Response to Reviewer FLBY**
>
> # 1. Evaluation Bias and Comparability of Goal Success Rates
>
> > **Q1**: The paper claims DigiData’s trajectories achieve 94.6% goal success (before filtering) compared to AitW’s 84%, but both were verified under the authors’ own system. Could this introduce evaluation bias or unfairness? How do the authors ensure comparability, and could a case study or qualitative inspection clarify what “quality difference” actually looks like?
>
> We took several steps to minimize evaluation bias and ensure fair comparability:
>
> - **Separate Workforce**: Our verification workforce and tooling is completely separate from our trajectory collection workforce and tooling. This way the annotators aren’t biased to they way they collect trajectories or execute tasks.
> - **Single-Blind Annotation**: Our annotation protocol is single-blinded—annotators are unaware of the source of each trajectory (DigiData or AitW). This prevents source-based bias in their assessments.
> - **Unified Verification Pipeline**: Both datasets were evaluated using the same verification pipeline and criteria, ensuring consistency in success rate measurement.
>
> We are open to sharing our annotation guidelines and sample annotated trajectories to further clarify the nature of “quality difference.”
>
> ---
>
> # 2. Motivation: Gaps in AitW and AndroidControl
>
> > **Q2**: What concrete gaps or shortcomings in AitW and AndroidControl specifically motivated the creation of DigiData?
>
> DigiData was created to address several specific limitations in existing datasets:
>
> - **Quality**: AitW’s lower trajectory quality was confirmed through our unified verification process. Many AitW trajectories fail to meet the standards required for robust agent training.
> - **Scale and Coverage**: AndroidControl is too small for large-scale learning. AitW, while larger, suffers from significant imbalance—87% of its data comes from Google Apps, and over half of its “apps” are actually websites, not native Android apps. In DigiData, website tasks are consistently categorized under Chrome, ensuring clarity and consistency.
> - **Task Diversity**: AitW’s Install Task category includes broad, high-level instructions (e.g., install, uninstall, login) that map to only three distinct tasks in DigiData, limiting diversity.
> - **Goal Enumeration**: Existing datasets often take a ‘breadth-first approach’: they provide only a handful of tasks per app while targeting a large number of apps. This can potentially create repetition and lack of coverage: the most obvious features are repeatedly scaled up across different apps, while the long tail of less popular features is never enumerated. DigiData takes a different approach by exhaustively enumerating all possible features and tasks for representative apps, which yields deeper coverage of the feature/task space, resulting in a more balanced and diverse dataset. Researchers can then leverage this advantage to study questions such as: for two functionally similar apps, A and B, how well does my model perform if trained on A (thoroughly) and evaluated on B?
>
> ---
>
> # 3. LLM-as-a-Judge: Novelty and Implementation
>
> > **Q3**: Since the use of LLM-as-a-judge is not novel, how does this implementation differ from or improve upon prior research on automated trajectory evaluation?
>
> We agree that LLM-based judging is an emerging standard. We don’t claim to have a novel Judge setup but report details of the judge for completeness and to follow general good practices.
>
> ---
>
> # 4. Benchmarking Against Other LLM-Based Assessment Studies
>
> > **Q4**: Can the authors benchmark their LLM judging pipeline against other existing studies applying LLMs for mobile control assessment?
>
> To our knowledge, there are currently no published studies that directly benchmark LLM-based judges for mobile control tasks at the scale and diversity presented in DigiData. We welcome future comparative studies and are open to collaborating with other researchers to establish standardized benchmarks in this area.
>
> ---
>
> We appreciate the reviewer’s recognition of DigiData’s strengths in dataset quality, scale, and benchmarking. While methodological novelty is not the primary focus, DigiData’s structured pipeline, exhaustive goal enumeration, and reproducible evaluation protocols represent significant advances over existing resources. We hope our clarifications address your concerns and demonstrate the value of releasing DigiData training dataset and benchmark to the research community.

---

### Official Review · Reviewer_iMrZ · 2025-11-01

**Soundness:** 2
**Presentation:** 3
**Contribution:** 3
**Rating:** 4
**Confidence:** 4

**Summary:**

This paper proposes Digidata, a large-scale dataset designed for training mobile control agents. The trajectories in this dataset are constructed through goal collection, human demonstration, and trajectory verification assisted by both human and model evaluation. The goals in Digidata are collected by exploring comprehensive features of mobile apps, thereby enhancing their diversity. Furthermore, Digidata provides multiple input modalities, including UI trees and chain-of-thought data. The paper also introduces a benchmark named DigiData-Bench for dynamically evaluating mobile agents.

**Strengths:**

The dataset is large-scale and diversified, facilitating robust training and evaluation of mobile agents.

**Weaknesses:**

- In the trajectory verification process, which specific LLM is used as the judge? Does this method incur high costs?

- There is a lack of comparison with other state-of-the-art mobile agent methods, such as UI-TARS[1], UI-Genie[2], and the Mobile-Agent series[3].

- Digidata-bench is proposed to address the inaccuracies of step accuracy metrics in offline evaluation. However, several online dynamic evaluation benchmarks already exist (e.g., AndroidWorld[4], SPA-Bench[5], A3[6]), and methods like LLM-as-a-judge have also been employed in SPA-Bench and A3. What is the core difference of Digidata-bench compared to these existing dynamic evaluation methods?

- As a benchmark, Digidata-bench lacks evaluation results on advanced closed-source models (e.g., Claude, Gemini) as well as leading open-source models like UI-TARS, UI-Genie, and the Mobile-Agent series.

[1] Qin Y, Ye Y, Fang J, et al. Ui-tars: Pioneering automated gui interaction with native agents[J]. arXiv preprint arXiv:2501.12326, 2025.

[2] Xiao H, Wang G, Chai Y, et al. UI-Genie: A Self-Improving Approach for Iteratively Boosting MLLM-based Mobile GUI Agents[J]. arXiv preprint arXiv:2505.21496, 2025.

[3] Ye J, Zhang X, Xu H, et al. Mobile-agent-v3: Fundamental agents for gui automation[J]. arXiv preprint arXiv:2508.15144, 2025.

[4] Rawles C, Clinckemaillie S, Chang Y, et al. Androidworld: A dynamic benchmarking environment for autonomous agents[J]. arXiv preprint arXiv:2405.14573, 2024.

[5] Chen J, Yuen D, Xie B, et al. Spa-bench: A comprehensive benchmark for smartphone agent evaluation[C]//NeurIPS 2024 Workshop on Open-World Agents. 2024.

[6] Chai Y, Li H, Zhang J, et al. A3: Android agent arena for mobile gui agents[J]. arXiv preprint arXiv:2501.01149, 2025.

**Questions:**

- Will the training dataset be open-sourced?

- How does Digidata-bench fundamentally differ from existing dynamic evaluation benchmarks such as AndroidWorld, SPA-Bench, and A3?

- Can you provide missing comparisons with other state-of-the-art mobile agent methods, including UI-TARS, UI-Genie, and the Mobile-Agent series?

---

> ### Author Response · Authors · 2025-11-27
> **Response to Reviewer iMrZ**
>
> # 1. Trajectory Verification Process and LLM Judge Costs
> > **W1**: In the trajectory verification process, which specific LLM is used as the judge? Does this method incur high costs?
>
> **LLM Used:**
>
> We used Llama 3.2.
>
> **Cost Analysis:**
>
> The overall cost comprises two components: human verification cost and LLM verification cost. Human verification is significantly more expensive than LLM-based verification. To optimize efficiency and maintain quality, we developed a pipeline that combines both methods: the LLM initially evaluates the trajectories, and only those that do not pass the LLM's assessment are forwarded to human reviewers. Approximately 87% of trajectories were judged by LLMs, with the remaining 13% undergoing human verification to ensure quality and resolve ambiguous cases. This hybrid approach allows us to control costs while upholding high verification standards.
>
> For the LLM-based evaluation, the cost per trajectory is proportional to the number of steps (N) in the trajectory, as full verification requires (N + 1) Llama 3 API calls.
>
> In practice, for this dataset release, we further enhanced quality by verifying over 95% of the data with human reviewers. This rigorous process ensures the dataset's high quality, resulting in a nearly 100% success rate (compared to AitW's 84%).
>
> ---
>
> # 2. Open-Sourcing the Training Dataset
> > **Q1**: Will the training dataset be open-sourced?
>
> Yes our training dataset will be open-sourced along with our static test set and dynamic benchmark.
>
> ---
>
> # 3. Comparison with Existing Dynamic Evaluation Benchmarks
> > **Q2**: How does Digidata-bench fundamentally differ from existing dynamic evaluation benchmarks such as AndroidWorld, SPA-Bench, and A3?
>
> A break down of the fundamental differences between DigiData-Bench, Android World, and SPA-Bench/A3:
>
> |                | Real 3p Apps | Online, popular Apps | Deterministic state/fair comparison |
> |----------------|:------------:|:-------------------:|:-----------------------------------:|
> | **AndroidWorld**   |      ✓       |         𐄂         |                 ✓                   |
> | **SPA-Bench/A3**      |      ✓       |         ✓          |                 𐄂                 |
> | **DigiData-Bench** |      ✓       |         ✓          |                 ✓                   |
>
> **Key Points:**
> - **Online, Real-World, Popular Apps with High Ecological Validity**: DigiData-Bench focuses on online evaluation using real, popular third-party apps, thereby increasing ecological validity compared to benchmarks that use less popular or offline apps.
> - **Deterministic State & Fair Comparison**: The full evaluation suite, including task specifications, environment initial states, and the LLM judge, is almost entirely deterministic (noting that online apps inevitably contain some dynamic content). We provide detailed setup instructions and a fully specified LLM evaluation harness (including models and prompts), ensuring consistent environment states and enabling fair, reproducible comparisons across models.
>
> **Shortcomings of Existing Benchmarks:**
> - **Offline Benchmarks (e.g., AiTW, AdCtrl)**: Rely on step accuracy, which we argue is insufficient for robust evaluation.
> - **Online Benchmarks on Less Popular Apps (e.g., AndroidWorld)**: Suffer from limited real-world relevance.
> - **Online Benchmarks on Popular Apps (e.g., SPA-Bench/A3)**: Often lack standardized environment state initialization, leading to inconsistent and unfair comparisons.
>
> **How DigiData-Bench Addresses These Limitations:**
>
> DigiData-Bench overcomes these issues by combining online evaluation, real-world app selection, standardized environment initialization, and a transparent evaluation pipeline.
>
> ---
>
> # 4. Comparisons with State-of-the-Art Mobile Agent Methods
> > **Q3**: Can you provide missing comparisons with other state-of-the-art mobile agent methods, including UI-TARS, UI-Genie, and the Mobile-Agent series?
>
> We acknowledge the absence of direct comparisons with recent state-of-the-art methods such as UI-TARS, UI-Genie, and the Mobile-Agent series. The primary reason is the lack of available evaluation results for these models on DigiData-Bench at the time of submission. We are actively working to expand our benchmark coverage and will include these comparisons in future releases as more results become available.
>
> ---
>
> # 5. Additional Clarifications
>
> > **W3**: ...What is the core difference of Digidata-bench compared to these existing dynamic evaluation methods?
>
> **LLM-as-a-Judge in Other Benchmarks**: We don’t claim to have a novel LLM-as-a-Judge setup as other methods like SPA-Bench and A3 use one. DigiData-Bench does fully specify the evaluation harness and LLM Judge setup to ensure consistent, reproducible results.

---

> ### Author Response · Authors · 2025-11-27
> **Response to Reviewer iMrZ**
>
> We appreciate the reviewer's feedback and hope these clarifications address your concerns. DigiData and DigiData-Bench are designed to advance the field by providing a robust, open, and fair platform for training and evaluating mobile control agents. We look forward to further community engagement and collaboration.

---

### Author Response · Authors · 2025-12-03
**Response to all reviewers**

We thank the reviewer for their constructive feedback and willingness to reconsider their score. We have clarified all raised questions and addressed points of confusion. Additionally, we conducted further experiments and included the results below to address the reviewers' concerns.

---

### 1. Evaluation on an Important Related Dataset – GUIOdyssey
The GUIOdyssey dataset features longer interaction sequences (average 15.3 steps) than the proposed DigiData.
We evaluated our models on GUIOdyssey and compared their performance against models considered ‘zero-shot’ (i.e., not trained on GUIOdyssey) as reported in the GUIOdyssey paper. The results are shown below:

|                | GUIOdyssey High Level | GUIOdyssey Low Level |
|----------------|:--------------------:|:--------------------:|
| **GPT-4V**             |      13.49         |      41.28         |
| **GPT-4o**             |      13.19         |      42.71         |
| **Claude3.5-Sonnet**   |      15.80         |      34.18         |
| **InternVL2-Pro**      |      16.04         |      43.98         |
| **CogAgent**           |      13.95         |      30.44         |
| **SphAgent**           |      15.98         |      30.00         |
| **Ours 8B**            |   **36.53**        |   **52.94**        |
| **Ours 8B COT**        |      35.32         |      52.13         |

Key Findings:

- Our model achieved the highest scores on both GUIOdyssey high-level and low-level tasks, outperforming other models by a significant margin.
- SphAgent has a setup similar to ours for GUIOdyssey evaluation (model size: 7B vs. 8B; both trained on other mobile control datasets and tested on GUIOdyssey). Our model outperformed SphAgent by 128% on high-level and 76% on low-level tasks, indicating that training on our dataset develops superior mobile control capabilities that transfer to out-of-domain datasets. This demonstrates the high quality, diversity, and representativeness of our dataset.

---

### 2. Inclusion of PLM Un-Finetuned Version as Baseline

We evaluated the largest PLM-based model (not finetuned on DigiData), the 8B version, and tested step accuracy on DigiData-Bench. The accuracy was 1%, compared to 72.8% for the finetuned 8B PLM model. This suggests that our base model has virtually no mobile control capabilities, and training on our dataset significantly improves performance.

---

### 3. Results on DigiData-Bench-Auto

We report the results on DigiData-Bench-Auto for our trained models:

|                | Success Rate (DigiData-Bench-Auto) |
|----------------|:----------------------------------:|
| **PLM 1B**     |              24.4%                 |
| **PLM 3B**     |              29.8%                 |
| **PLM 8B**     |              35.1%                 |
| **PLM 8B COT** |              36.9%                 |

The trend aligns well with step accuracy and DigiData-Bench results.

---

### 4. Benchmarking More Leading Models on DigiData-Bench (-Auto)

At the time of submission, we benchmarked the most representative leading closed-source model (GPT-4o) and open-source model (Qwen2.5VL), along with our trained models as baselines. We are committed to conducting further experiments on recent mobile control agents on DigiData-Bench to cover more work. We will include additional results in the revised version and are committed to maintaining a leaderboard in the DigiData-Bench repository, which we believe is the best place to track recent advancements and encourage community engagement.

---

We believe these additions will further strengthen the submission and clarify the contributions of DigiData to the community.

---

### Author Response · Authors · 2025-12-03
**Summary for Area Chair**

Our paper introduces **DigiData**, a large-scale, high-quality, diverse, multi-modal dataset designed for training mobile control agents—a resource notably lacking in the community. We further propose **DigiData-Bench** and **DigiData-Bench-Auto**, benchmarks for evaluating mobile control agents on real-world, complex tasks. DigiData-Bench-Auto enables end-to-end evaluation on real-world, online popular top apps, supported by rigorous state initialization to ensure fairness. We conducted extensive experiments on agent models and LLM Judge models, which demonstrate strong performance gains from using DigiData and validate the effectiveness of LLM Judge in our DigiData-Bench.

---

## Strengths

Reviewers found our work:

- Well-motivated, valuable, and essential; it serves as a solid foundation for training and evaluating mobile control agents (iMrZ, FLBY, PvGF, Fk85).
- The dataset is large-scale (second largest), high-quality (~100% passing rate compared to 84% for the current largest dataset), and information-rich (screenshots, UI trees, reasoning), with high goal complexity and diversity. Detailed data construction protocols and analysis are included (iMrZ, FLBY, PvGF, Fk85).
- Experimental results are strong and clearly demonstrate performance gains from using DigiData, providing evidence supporting the use of LLM-as-a-Judge in DigiData-Bench (FLBY, PvGF).

---

## Weakness, concerns and our responses

We have clarified and addressed all reviewer questions and conducted additional experiments to further strengthen the paper.
Among the questions, a few notable ones include:

---

**Q:  How is the DigiData dataset different or better than existing datasets? (FLBY)**

**A**:  Given the scarcity of high-quality mobile control datasets, DigiData is the second largest and significantly higher in quality than the largest existing dataset, Android-in-the-Wild (AitW): ours achieves a 100% passing rate versus AitW’s 84%. DigiData also offers greater diversity and comprehensive coverage of all reasonable features and tasks across a broad range of app types.

---

**Q: How does DigiData-Bench differ from existing benchmarks? (iMrZ)**

**A**: DigiData-Bench is the first mobile control benchmark to meet all of the following requirements: high ecological validity (i.e., real-world, online popular apps), deterministic state initialization for fair comparison, and a complete end-to-end evaluation pipeline.

---

**Q: LLM Judge is incremental, lacking novelty and comprehensive comparison against others (iMrZ, FLBY)**

**A**: Novelty is not the primary contribution we claim for LLM Judge. Our contribution is the integration of LLM Judge into our end-to-end benchmark, enabling comprehensive evaluation. We also compare LLM-based evaluation against human evaluation, providing detailed analysis and justifying its use as a reasonable proxy for end-to-end assessment.

---

**Q: Availability and qualitative analysis of DigiData dataset quality and diversity (iMrZ, Fk85)**

**A**: We have included a Google Colab notebook (linked on the first page) that allows reviewers to download the full dataset and provides tools for spot-checking the claimed quality in the most straightforward way. The dataset and benchmark will be fully open-sourced. We encourage reviewers and area chairs to explore the Colab notebook and anonymous GitHub repository to examine and appreciate the strengths of our dataset and benchmark.

---

### Meta-Review · Area_Chair_fQNn · 2026-01-04

**Summary:**

This papers receives reviews with score 4. The main concerns are limited novelty, insufficient sota comparisons, and unclear positioning differs from current dynamic benchmarks. I generally think these concerns are reasonable and cannot be easily addressed during the rebuttal.

**Reviewer Concerns:**

The remaining concerns about the papers would still be:


1. Limited novelty / incremental contributions: dataset + benchmark seen as mostly scale/engineering; LLM-as-a-judge viewed as incremental and not a core novelty.

2. Insufficient SOTA comparisons: missing evals vs leading mobile agents (e.g., UI-TARS, UI-Genie, Mobile-Agent series, OdysseyAgent/GUI-Owl) and more closed-source models (e.g., Claude, Gemini).

3. Unclear differentiation vs existing dynamic benchmarks: reviewers asked how DigiData-Bench fundamentally differs from AndroidWorld, SPA-Bench, A3, especially since LLM-judge exists elsewhere.

**Reviewer Scores:**

I do not think the rebuttal could change the all negative scores to the scores making the paper get accepted.

---

### Decision · Program_Chairs · 2026-01-26

Reject